# Learning Dynamics of VLM Finetuning: Cooling-Weighted DPO with Mixed Negatives

## Abstract

Preference-based finetuning of vision–language models (VLMs) is brittle: trivially wrong negatives inject uninformative gradients that destabilize training. We recast alignment as **learning-dynamics–aware optimization** and introduce **Cooling-Weighted DPO (CW-DPO)**, a two-stage recipe that explicitly models and exploits the training trajectory. **Stage 1** performs supervised finetuning with **gentle negatives**: **low-weight smoothed supervision** that regularizes the base policy and curbs overconfidence without explicit penalties. **Stage 2** applies a DPO objective in which the **negative term is scaled by a cooling weight** computed from the model's **average token log-probability** on each negative, suppressing uninformative gradients from easy or off-distribution samples while preserving signal from hard negatives. In practice, we emphasize **on-policy negatives** and allow **mixed negatives** by blending a controllable fraction of dataset negatives to maintain contrast freshness. Throughout, we instrument training with $\Delta \log p$ probes on positives and negatives as first-class signals for early stopping, curriculum design, and failure diagnosis. Across diverse VLM tasks, CW-DPO yields **more stable optimization**, **better calibration**, and **higher pairwise win-rates** than SFT-only and vanilla DPO, while **converging in fewer steps**. Ablations isolate the **cooling-weight mechanism** as the primary driver of these gains and show complementary benefits from mixing on-policy and dataset negatives. Taken together, our results show that **smoothing learning dynamics before cooling preferences** is a simple, general principle for robust VLM alignment.

## 1 INTRODUCTION

The finetuning of vision-language models (VLMs) involves intricate learning dynamics that pose significant challenges for stable optimization (Liu et al., 2023; Huang & Zhang, 2024). VLMs process multimodal inputs, encoding textual and visual components as high-dimensional sequences, where the visual stream introduces complex state dependencies—such as pixel embeddings and spatial metadata—that tightly couple gradient updates across tokens (Radford et al., 2021; Li et al., 2023). Prominent finetuning methods, including supervised finetuning (SFT) (Ouyang et al., 2022) and direct preference optimization (DPO) (Rafailov et al., 2023), employ diverse loss geometries and supervision signals, necessitating a unified analytical framework to unravel their behavioral foundations, especially in preference-based alignment aimed at prioritizing human-preferred outputs (Ren & Sutherland, 2025). Preference-based finetuning is essential for aligning VLMs with human intent (Liu et al., 2024a; Radford et al., 2021; Chen et al., 2023; Zhang et al., 2024), yet it suffers from notorious instability in practice. Alignment datasets often contain static or mis-specified negative examples—trivially incorrect or off-distribution—that inject uninformative gradients (Casper et al., 2023; Kaufmann et al., 2024; Song et al., 2025). These gradients disrupt optimization, degrade calibration, and produce overconfident, peaky posteriors. Off-policy methods exacerbate this by penalizing unlikely responses, while even naïve on-policy approaches struggle with gradient spikes from dominant "easy negatives" (Christiano et al., 2017; Kaufmann et al., 2024). This points to a common flaw: alignment is often

treated as a static optimization task, ignoring the dynamic evolution of the model's learning trajectory (Ren & Sutherland, 2025; Gao et al., 2023; Kaufmann et al., 2024).

In this work, we adopt a learning-dynamics perspective, reframing alignment to explicitly model and harness how the model's beliefs evolve during finetuning (Sagawa et al., 2020). We introduce Cooling-Weighted Direct Preference Optimization (CW-DPO), a two-stage strategy that aligns with this evolution. The first stage smooths the loss landscape to enhance stability, while the second applies a competence-aware preference optimization to refine training, as depicted in Figure 2. Specifically, Stage 1 enhances SFT by incorporating "gentle negatives," introducing low-weight smoothed supervision to reduce over-confidence around negative responses without harsh penalties. We define the per-token average log-probability as $\bar{\ell}_\theta(y \mid \chi) = \frac{1}{L} \sum_{l=1}^{L} \log \pi_\theta(y_l \mid \chi_{\leq l})$, measuring the model's average confidence per token on any response $y$ given sample $\chi$ (elaborated in §2.2), with $y_l$ (loser) and $y_w$ (winner) specifying roles in Stage 2. The objective, formalized as a constrained optimization (detailed in §3), is: $\min_\theta \mathbb{E}_{(x,y^+)\sim\mathcal{D}}[-\log \pi_\theta(y^+ \mid x)] + \eta \mathcal{R}_{\text{smooth}}(\theta; x, y^-)$, $0 < \eta \ll 1$, where $\mathcal{R}_{\text{smooth}}$ (e.g., entropy smoothing or a ReLU-based soft constraint) regularizes the negative trajectory $y^-$. This "smooth-before-optimize" approach de-peaks distributions and flattens sharp loss regions, reducing noise in subsequent contrastive learning, as motivated by the peaking pitfalls in §2.1. In Stage 2, we transition to preference pairs $y_w$ (winner) and $y_l$ (loser), as detailed in §3. Stage 2 advances with a novel DPO-style objective featuring competence-aware reweighting (Rafailov et al., 2023). Vanilla DPO minimizes $-\log \sigma(\beta(\Delta_w - \Delta_l))$, where $\Delta_w = \log \pi_\theta(y_w \mid x) - \log \pi_{\text{ref}}(y_w \mid x)$ and $\Delta_l = \log \pi_\theta(y_l \mid x) - \log \pi_{\text{ref}}(y_l \mid x)$. We enhance it with a cooling weight: $w_c(\theta; y_l, \chi) = \sigma\left(\frac{\bar{\ell}_\theta(y_l \mid \chi) - \ell_{\text{floor}}}{\tau}\right)$, which down-weights $y_l$ with low probabilities (indicating "easy" negatives), steering optimization toward hard negatives where uncertainty lingers. The resulting loss is: $\mathcal{L}_{\text{CW-DPO}} = -\mathbb{E}\left[\log \sigma(\beta(\Delta_w - w_c(\theta; y_l, \chi) \cdot \Delta_l))\right]$, where $\ell_{\text{floor}}$ sets an easiness baseline and $\tau$ adjusts the cooling schedule's sharpness. Negatives are primarily on-policy, with optional dataset-negative mixing to keep contrast fresh. Across both stages, $\Delta \log p$ probes on a held-out set monitor learning dynamics, providing a low-cost signal for early stopping and curriculum design. This endogenous curriculum adapts to model competence. Extensive and comprehensive experimental evaluations in §4 demonstrate that our CW-DPO surpasses SFT-only and vanilla DPO in stability, efficiency, calibration, and win-rates across visual QA, binary judgments, and open-ended tasks.

## 2 PROBLEM FORMULATION: THE UNSTABLE DYNAMICS OF VLM FINETUNING

We systematically dissect the core instabilities afflicting VLM alignment, i.e., **a fundamental dilemma in preference-based learning, manifesting as the "squeezing effect,"** in §2.1. This effect underscores **a perilous decoupling between a sample's loss-based informativeness and its gradient-based influence** during training. Subsequently, in §2.2, we develop a formal analytical lens rooted in learning dynamics to diagnose this issue. This framework not only elucidates the root causes of instability but also **yields a principled blueprint for our dynamics-aware solution**.

### 2.1 A CORE DILEMMA: THE "SQUEEZING EFFECT"

The **fundamental dilemma** of preference finetuning is that aligning with human intent requires penalizing a vast space of undesirable responses ($y^-$) (Kaufmann et al., 2024). As learning progresses, most undesirable responses are gradually converted into "easy negatives", i.e., sequences assigned near-zero probability by the model. This engenders **a destructive feedback loop**, wherein optimization expends disproportionate gradient bandwidth on these uninformative samples. As shown in Figure 1, the consequence is the **squeezing effect**, i.e., a decoupling where a sample's low loss (indicating minimal informativeness) belies its potentially large, misdirected gradient (Ren & Sutherland, 2025). Although the loss from an easy negative $\pi_\theta(y^- \mid x) \to 0$ is negligible, its gradient can remain substantial and poorly aligned. This misalignment in-

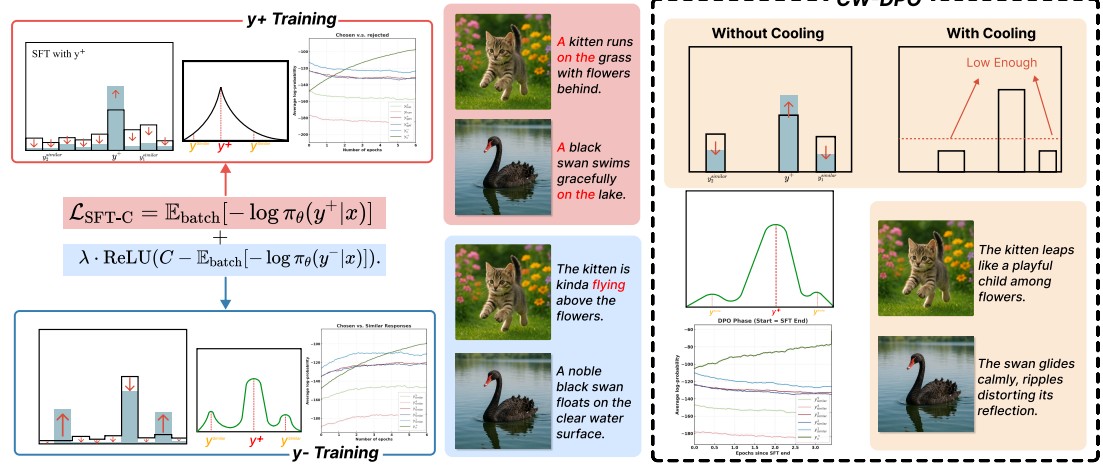

Figure 1: Two-stage optimization process of CW-DPO. Stage 1 ($y^+$ Training) leverages positive supervision for stability but yields overly uniform language styles (e.g., "A ... on the ..."). Stage 2 ($y^-$ Training) introduces negative contrast for variation but risks errors (e.g., a running kitten as "flying"). CW-DPO's cooling-weighted mechanism dynamically attenuates uninformative negatives while amplifying hard ones, mitigating error propagation, and enhancing stylistic diversity.

duces an undesirable redistribution of probability mass: instead of fostering a calibrated spread across viable alternatives, updates "squeeze" mass toward the dominant mode, typically $y^* = \arg\max_y \pi_\theta(y|x)$, which may correspond to a preferred response $y_w$ in later optimization stages. **This engenders a "rich-get-richer" dynamic**, amplifying overconfidence, curtailing linguistic diversity, and impairing calibration.

**Remark 1** (Insufficiency of DPO's Implicit Regularization). *DPO implicitly counters this via regularization: the negative-term gradient is modulated by $\beta(1-a)$, where $a = \sigma(\beta(\Delta_w - \Delta_l))$ is the sigmoid-transformed margin. For extremely easy negatives, $\Delta_l$ drives $a \to 1$, attenuating the gradient. Theoretically elegant, this falters in practice due to a wide "vulnerable region" for **moderately easy negatives**, where $\log \pi_\theta(y^-)$ is low but $a$ (e.g., $\in [0.8, 0.99]$) insufficiently suppresses the residual gradient $\beta(1-a)$, especially at high $\beta$ (Ren & Sutherland, 2025). This perpetuates instability and the squeezing effect. (See Appendix I for a formal analysis).*

## 2.2 An Analytical Lens: Per-Step Influence Decomposition

To transcend empirical observations and rigorously diagnose the squeezing effect, we adopt a learning-dynamics perspective (Koh & Liang, 2017; Jacot et al., 2018) to enable precise tracing of how a single gradient update impacts global model behavior. Define $y = (y_1, \ldots, y_L)$ as a sequence of length $L$, with logits $z = (z_1, \ldots, z_L)$, each $z_l \in \mathbb{R}^{|V|}$ ($|V|$ denotes the vocabulary size). Gradients are w.r.t. the concatenated $z$, denoted $\nabla_z$. A pivotal query: How does an update on "updating" sample $\chi_u = (x_u, y_w, y_l)$ alter confidence on "observing" sample $\chi_o$? Confidence is quantified via average per-token log-probability: $\bar{\ell}_\theta(y \mid \chi) = \frac{1}{L} \sum_{l=1}^{L} \log \pi_\theta(y_l \mid \chi_{\leq l})$. A first-order Taylor expansion of $\bar{\ell}_\theta$ post-update $\theta_{t+1} = \theta_t - \eta \nabla_\theta \mathcal{L}(\theta_t; \chi_u)$ yields:

$$\Delta \bar{\ell}_t(y|\chi_o) = -\eta (\nabla_\theta \bar{\ell}_{\theta_t})^\top (\nabla_\theta \mathcal{L}(\theta_t)) + \mathcal{O}(\eta^2). \tag{1}$$

Linearizing logits $z(\theta; \chi)$ around $\theta_t$ decomposes this into interpretable factors.

**Proposition 1** (Sequence-Aware One-Step Influence). *The log-likelihood change on $\chi_o$ post-update on $\chi_u$ (rate $\eta$) approximates:*

$$\Delta\bar{\ell}_t(y \mid \chi_o) \approx -\eta \big\langle \underbrace{\nabla_z\bar{\ell}_{\theta_t}(y \mid \chi_o)}_{A_t: \text{ Belief Geometry}}, \underbrace{K_t(\chi_o, \chi_u)}_{\text{eNTK Kernel}} \underbrace{\nabla_z\mathcal{L}(\theta_t; \chi_u)}_{G_t: \text{ Loss Residual}} \big\rangle. \tag{2}$$

*Key Elements: **Belief Geometry** ($A_t$) encodes predictive sensitivity to logit perturbations, capturing belief-landscape curvature. **eNTK Kernel** ($K_t = J_o J_u^\top$) ($J = \nabla_\theta z(\theta_t; \chi)$: Jacobian) propagates updates parametrically. **Loss Residual** ($G_t$) directs logit adjustments via $\nabla_z\mathcal{L}$.*

**Decomposing the DPO Gradient. The power of this decomposition becomes evident when we specify the Loss Residual $G_t$ for the DPO objective.** For DPO, $G_t = \nabla_z\mathcal{L}_{\text{DPO}}$ (derived in Appendix I), whose full form is given in Eq. 5 and can be broken down into components related to the winner $y_w$ and the loser $y_l$: $G_t = \beta(1-a)(G_t^w - G_t^l)$, where $G_t^w$ and $G_t^l$ are the gradient components for the winning and losing responses, respectively. As discussed in Remark 1, the squeezing effect occurs precisely when $y_l$ is an "easy negative." In this scenario, while the loss itself is small, DPO's implicit regularization $(1-a)$ is often insufficient to fully suppress the gradient, leaving the loser component $G_t^l$ disproportionately large and noisy. This oversized residual from uninformative samples is the direct source of instability.

**Implication for Algorithm Design.** This analysis transcends explanation: **it isolates the instability's source to the oversized, destabilizing "loser" component ($G_t^l$) of the loss residual from negative examples** $y_l$. The squeezing effect, therefore, emerges not from an inherent flaw in preference optimization but from an unregulated $G_t^l$. **This mandates a surgical solution**: instead of heuristically regularizing the entire loss, a principled algorithm must directly temper this specific residual component. This diagnosis is the analytical foundation for our method, detailed in the next section 3.

## 3 DYNAMICS-AWARE COOLING-WEIGHTED DPO

Grounded in the principled insights of our diagnostic analysis (§2), our CW-DPO in Figure 2 provides a dynamics-aware manner to align VLMs.

### 3.1 STAGE 1: TRAJECTORY PRIMING VIA CONSTRAINED SFT

This stage prepares the learning trajectory of the model $\pi_\theta$ ($\theta$ denotes the model parameters) by curbing overconfidence, laying a smoother foundation for subsequent preference learning. Unlike standard SFT, which focuses solely on positive responses ($y^+$) and risks entrenching peaky distributions, we adopt a constrained optimization strategy. To mitigate overconfidence, we impose a constraint on the model's response to negatives, minimizing the negative log-likelihood (NLL) on positives while ensuring the NLL on negatives ($y^-$) remains above a threshold $C$ to prevent their premature dismissal as:

$$\min_\theta \mathbb{E}_{(x,y^+)\sim\mathcal{D}}[-\log\pi_\theta(y^+|x)] \quad \text{s.t.} \quad \mathbb{E}_{(x,y^-)\sim\mathcal{D}}[-\log\pi_\theta(y^-|x)] \geq C. \tag{3}$$

Here, the objective $\mathbb{E}_{(x,y^+)\sim\mathcal{D}}[-\log\pi_\theta(y^+|x)]$ seeks to maximize the likelihood of positive responses $y^+$ drawn from dataset $\mathcal{D}$, while the constraint $\mathbb{E}_{(x,y^-)\sim\mathcal{D}}[-\log\pi_\theta(y^-|x)] \geq C$ ensures that the model assigns sufficient probability to negative examples $y^-$, preventing them from being overly suppressed. This dual focus promotes a more uniform allocation of probability mass, countering the peaking pitfalls outlined in §2.1, where overconfidence on easy negatives distorts the loss landscape. To solve this constrained problem practically, we apply a Lagrangian relaxation, introducing a penalty term to softly approximate the constraint. This leads to the Smoothed SFT loss:

$$\mathcal{L}_{\text{SFT-C}} = \mathbb{E}_{\text{batch}}[-\log\pi_\theta(y^+|x)] + \lambda \cdot \text{ReLU}(C - \mathbb{E}_{\text{batch}}[-\log\pi_\theta(y^-|x)]). \tag{4}$$

The first term, $\mathbb{E}_{\text{batch}}[-\log\pi_\theta(y^+|x)]$, remains the standard NLL for positive examples, computed over minibatches for efficiency. The second term, $\lambda \cdot \text{ReLU}(C - \mathbb{E}_{\text{batch}}[-\log\pi_\theta(y^-|x)])$, acts as a regularization: if

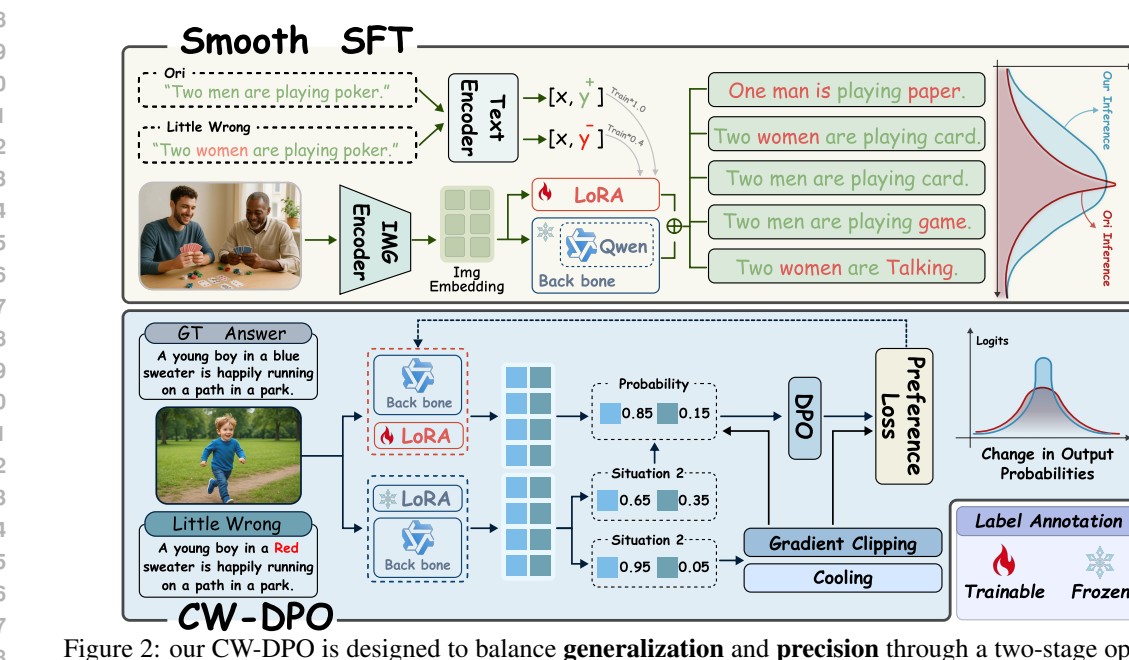

Figure 2: our CW-DPO is designed to balance **generalization** and **precision** through a two-stage optimization strategy. In Stage 1, *Smooth SFT* leverages positive samples together with negative samples containing minor errors to construct a smoothed supervision signal. This broadens the model's output probability distribution, thereby enhancing its generalization ability and robustness. In Stage 2, our CW-DPO employs preference pairs with fine-grained errors for DPO. By sharpening the probability distribution, this stage strengthens the model's capacity for precise discrimination of critical details.

the expected NLL of negatives falls below $C$, the ReLU activates, penalizing the model with a strength proportional to $\lambda$. This soft enforcement encourages the model to maintain a balanced response to negatives without rigid enforcement, approximating the original constraint stochastically. Here, mini-batch expectations provide practical approximations, and the ReLU term gently nudges the model toward a well-calibrated initialization. This process stabilizes the Belief Geometry ($A_t$ in Prop. 1), setting the stage for the targeted adjustments in Stage 2 by smoothing the initial loss landscape.

### 3.2 STAGE 2: COMPETENCE-AWARE PREFERENCE OPTIMIZATION

§2.2 reveals that instability stems from gradient updates for the negative (loser) sample $y_l$, particularly the loser component of the Loss Residual ($\tilde{G}_t$), which generates oversized and uninformative updates for easy negatives. By asymmetrically applying a cooling weight $w_c$ to the loser's log-probability difference $\Delta_l$, we achieve precise control over gradient influence. **Vanilla DPO Gradient.** Consider the DPO loss for a preference pair $(y_w, y_l)$: $\mathcal{L}_{\text{DPO}} = -\log \sigma\big(\beta(\Delta_w - \Delta_l)\big)$, where $\Delta_{w/l} = \log \pi_\theta(y_{w/l}|x) - \log \pi_{\text{ref}}(y_{w/l}|x)$. The gradient with respect to the logits (the Loss Residual) is:

$$G_t^{\text{DPO}} = \nabla_z \mathcal{L}_{\text{DPO}} = \beta(1 - a)\left((g_w - g_{\text{ref}}^w) - (g_l - g_{\text{ref}}^l)\right), \tag{5}$$

where $a = \sigma(\beta(\Delta_w - \Delta_l))$ and $g_{w/l} = \nabla_z \log \pi_\theta(y_{w/l}|x)$. Our decomposition pinpoints the squeezing effect to the loser term $(g_l - g_{\text{ref}}^l)$, which drives instability for easy negatives (Ren & Sutherland, 2025).

**Cooling Weight: Principled Modulator.** To address this, we introduce the cooling weight $w_c$, which adjusts the negative-sample gradient based on real-time model confidence:

$$w_c(\theta; y_l, \chi) = \sigma\left(\frac{\bar{\ell}_\theta(y_l \mid \chi) - \ell_{\text{floor}}}{\tau}\right), \tag{6}$$

**Algorithm 1** The Two-Stage CW-DPO Finetuning Protocol

**Input:** Pretrained VLM $\theta_0$, dataset $\mathcal{D}$, hyperparameters $\lambda, C, \beta, \tau, \ell_{\text{floor}}$, learning rate $\alpha$.
**Output:** Finetuned VLM parameters $\theta$.
1: **Initialize:** Policy model $\theta \leftarrow \theta_0$.

  *Stage 1: Trajectory Priming*
2: **for** $t = 1, \ldots, T_1$ **do**
3:   Sample a mini-batch $(x, y^+, y^-)$ from $\mathcal{D}$.
4:   Compute Smoothed SFT loss $\mathcal{L}_{\text{SFT-C}}$ using Eq. 4.
5:   Update parameters: $\theta \leftarrow \theta - \alpha \nabla_\theta \mathcal{L}_{\text{SFT-C}}$.
6: **end for**
7: Set reference model: $\pi_{\text{ref}} \leftarrow \pi_\theta$.

  *Stage 2: Cooled Preference Optimization*
8: **for** $t = 1, \ldots, T_2$ **do**
9:   Sample a mini-batch of preferences $(x, y_w, y_l)$ from $\mathcal{D}$.
10:   Compute cooling weight $w_c$ for each sample using Eq. 6.
11:   Compute CW-DPO loss $\mathcal{L}_{\text{CW-DPO}}$ using Eq. 7.
12:   Update parameters: $\theta \leftarrow \theta - \alpha \nabla_\theta \mathcal{L}_{\text{CW-DPO}}$.
13: **end for**

14: **return** Finetuned parameters $\theta$.

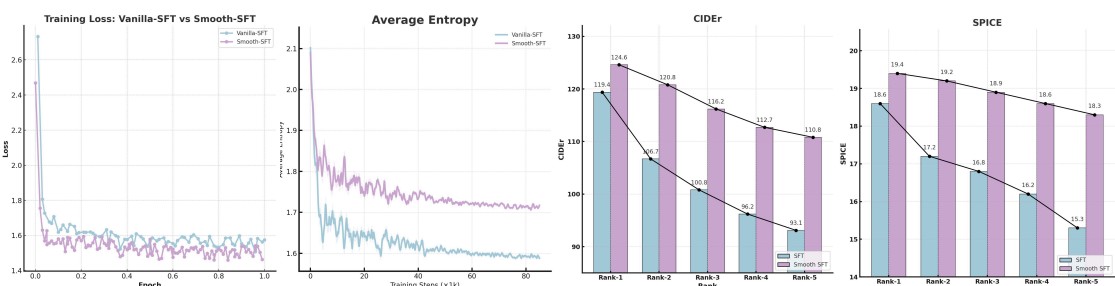

Figure 3: Validation of Stage 1 Constrained SFT (SFT-C) vs. standard SFT on: (1) loss; (2) entropy; (3) CIDEr; and (4) SPICE for Top-5 generations. SFT-C sustains higher entropy (less squeezing) and quality.

where $\bar{\ell}_\theta(y_l \mid \chi)$ is the average per-token log-probability (as defined in §2.2), $\ell_{\text{floor}}$ establishes an "easiness" baseline, and $\tau$ controls the transition sharpness, with higher values yielding a smoother weighting. For confidently rejected responses ($\bar{\ell}_\theta \ll \ell_{\text{floor}}$), $w_c \to 0$, nullifying the gradient; for uncertain hard negatives ($\bar{\ell}_\theta \geq \ell_{\text{floor}}$), $w_c \to 1$, preserving the learning signal.

**Core Loss Function.** We integrate $w_c$ asymmetrically, dampening only $\Delta_l$, to define our core loss:

$$\mathcal{L}_{\text{CW-DPO}} = -\log \sigma \left( \beta \left( \Delta_w - w_c(\theta; y_l, \chi) \cdot \Delta_l \right) \right), \tag{7}$$

Differentiating (treating $w_c$ as locally constant) yields the cooled residual $G_t^{\text{CW}} = \nabla_z \mathcal{L}_{\text{CW-DPO}}$ as:

$$\nabla_z \mathcal{L}_{\text{CW-DPO}} = \beta(1 - a') \left( \nabla_z \Delta_w - w_c \nabla_z \Delta_l \right) = \beta(1 - a') \left( (\pi_\theta(\cdot|x) - y_w) - w_c \cdot (\pi_\theta(\cdot|x) - y_l) \right),$$

where $a' = \sigma(\beta(\Delta_w - w_c \Delta_l))$. This $G_t^{\text{CW}}$ ensures gradients from easy negatives are minimized, preserving positive updates, and resolves the squeezing effect for stable, superior alignment. See Algorithm 1 for the full protocol. In essence, CW-DPO stabilizes training by smoothing initial losses and refining preferences with competence-aware weights, as validated empirically in §4.

Table 1: Performance comparison on vision-language benchmarks. For COCO, Flickr30k, and NoCaps, we report BLEU-4 (B@4), METEOR (M), CIDEr (C), and SPICE (S), with NoCaps split into In, Near, Out, and Entire. We also report accuracy on MMMU and MMBench1.1. Best results are in **bold**.

| Method | COCO Test | | | | Flickr30k Test | | NoCaps Val | | | | MMMU | MMBench |
|---|---|---|---|---|---|---|---|---|---|---|---|---|
| | B@4 | M | C | S | C | S | In | Near | Out | Entire | | |
| Qwen2.5-VL (Base) | 31.2 | 26.2 | 128.8 | 23.8 | 78.9 | 17.2 | 115.6 | 113.7 | 117.6 | 116.2 | 70.2 | 84.9 |
| SFT | 35.2 | 28.4 | 136.5 | 24.3 | 83.2 | 17.5 | 121.2 | 118.5 | 120.1 | 120.4 | 71.8 | 86.2 |
| DPO | 33.5 | 28.0 | 136.9 | 24.0 | 86.5 | 18.0 | 119.5 | 117.2 | 119.8 | 118.9 | 71.1 | 84.9 |
| PPO | 34.9 | 28.7 | 139.2 | 24.7 | 82.1 | 17.7 | 120.2 | 118.9 | 120.0 | 119.7 | 71.4 | 85.8 |
| V-DPO | 36.6 | 28.7 | 138.3 | 24.8 | 86.3 | 18.2 | 122.5 | 119.0 | 121.6 | 121.0 | 72.9 | 86.8 |
| GRPO | 36.5 | 28.8 | 138.2 | 24.9 | 86.4 | 18.1 | 122.3 | 119.1 | 121.5 | 120.9 | 72.8 | 86.9 |
| OPA-DPO | 36.8 | 29.0 | 138.5 | 25.1 | 86.7 | 18.2 | 122.6 | 119.4 | 121.8 | 121.3 | 73.1 | 87.2 |
| **CW-DPO (Ours)** | **39.6** | **30.4** | **142.6** | **25.8** | **89.2** | **18.6** | **125.6** | **121.3** | **123.7** | **123.6** | **74.6** | **89.6** |

## 4 EXPERIMENTS

### 4.1 MAIN RESULTS ON STANDARD BENCHMARKS

We evaluate our CW-DPO on three standard image captioning benchmarks: COCO (Lin et al., 2015), Flickr30k (Young et al., 2014), and NoCaps (Agrawal et al., 2019) for generalization assessment, as well as two comprehensive multi-task evaluation benchmarks: MMMU (Yue et al., 2024) and MMBench (Liu et al., 2024c). For COCO and Flickr30k, we adopt the widely used Karpathy split. The backbone for our CW-DPO in all the experiments is **Qwen2.5-VL-72B** (Bai et al., 2025), and we compare it against a series of strong fine-tuning baselines, including SFT (Ouyang et al., 2022), vanilla DPO (Rafailov et al., 2023), PPO (Schulman et al., 2017), and GRPO (Shao et al., 2024). To ensure robustness, all reported results are averaged over **five independent runs**. As for **training protocol**, our CW-DPO follows a two-stage paradigm, i.e., Constrained SFT on 75% of the data and Preference Alignment on the remaining 25%. In Stage 2, preference pairs are built by synthesizing minimally perturbed alternatives $y_l$ for each winning caption $y_w$ via GPT-4o.

In Table 1, our CW-DPO consistently outperforms all compared methods, including recent DPO variants like V-DPO (Xie et al., 2024b), GRPO, and OPA-DPO (Yang et al., 2025), across 5 mainstream vision-language benchmarks. On COCO Test, our CW-DPO achieves a new SOTA CIDEr score of **142.6**, surpassing the strongest baseline PPO by 3.4 points (+2.4%). It also yields a high BLEU-4 score of **39.6**, marking a substantial improvement of 2.8 points (+7.6%) over OPA-DPO, reflecting enhanced overall generation quality. On Flickr30k Test that evaluates cross-domain generalization, our CW-DPO continues to lead all baselines with a CIDEr score of **89.2**, 2.5 points higher than the next-best method, OPA-DPO. This suggests that the training stability introduced by our CW-DPO translates effectively into stronger generalization across distribution shifts. On the more challenging NoCaps, our CW-DPO achieves leading performance across all subsets with an overall score of **123.6**. Notably, the gain on the out-of-domain split (+1.9) does not come at the expense of in-domain performance (+3.0) when compared to the strongest baselines, indicating a favorable trade-off between generalization and retention of core knowledge. Furthermore, CW-DPO achieves strong adaptability on two multi-task evaluation suites by obtaining an accuracy of **74.6%** on MMMU, outperforming the strongest baseline OPA-DPO (73.1%) and attaining the highest accuracy of **89.6%** on MMBench. This confirms our CW-DPO extends beyond captioning to broader multimodal reasoning tasks. Note that, vanilla DPO underperforms SFT on lexical metrics such as BLEU-4. This justifies our **core hypothesis** that naive preference optimization over easy negatives may induce over-penalization, thereby degrading generation quality.

### 4.2 PHASE-ONE SMOOTHING VALIDATION EXPERIMENT

To isolate and verify the effectiveness of our Constrained SFT (SFT-C) in mitigating the squeezing effect, we conduct a targeted validation experiment. We train two models on 75% of the COCO training data (∼85k

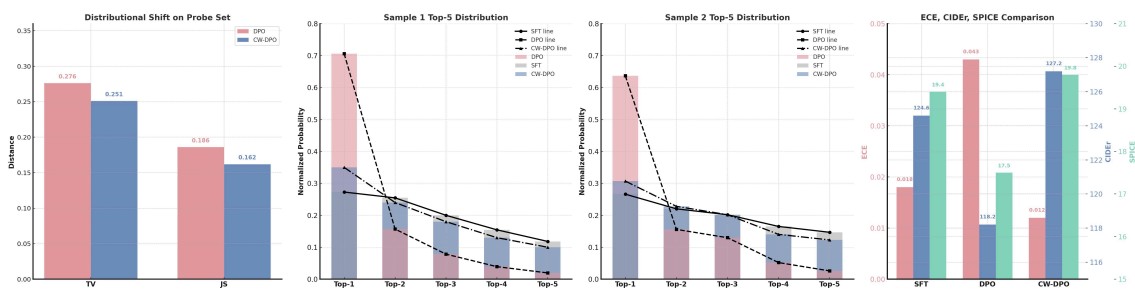

Figure 4: CW-DPO alleviates the squeezing effect of vanilla DPO, yielding smaller distribution shifts (left), smoother posteriors (middle), and improved generation quality with better calibration (right).

samples): one with standard SFT and the other with SFT-C. During training, both models are periodically evaluated on a fixed *probe set* of 1,000 examples from the COCO validation split. To quantify the squeezing effect, we measure the **average entropy** of the model's predictive distribution over the probe set. Lower entropy signifies a sharper, less diverse ("squeezed") distribution. To ensure the increased smoothness does not degrade generation quality, we also compute the **CIDEr** and **SPICE** scores for the **Top-5** predictions of each model at every evaluation step. Figure 3 validates the advantage of our SFT-C. The standard SFT exhibits a rapid decrease in training loss, but this is coupled with a **precipitous drop in predictive entropy**. This confirms that standard SFT quickly develops an overconfident, peaky distribution, i.e., a clear indicator of the squeezing effect, when overfit to the training data's dominant patterns. In contrast, SFT-C successfully maintains a **significantly higher entropy** throughout training, preserving predictive diversity. The slightly higher training loss observed for SFT-C is not a sign of inferior learning but rather an indication that the model is actively avoiding collapse into a narrow mode, resulting in a smoother, more generalized distribution. Crucially, this enhanced smoothness directly translates to superior generation quality. The **sustained higher CIDEr and SPICE scores** for SFT-C (Figure 3, right panels) demonstrate that by preventing the distribution from becoming overly sharp, our CW-DPO explores a richer semantic space, consistently producing more accurate and diverse top-k candidates.

### 4.3 PHASE-TWP: QUANTITATIVE ANALYSIS OF SQUEEZING EFFECT SUPPRESSION

To evaluate the effectiveness of our CW-DPO in mitigating the "squeezing effect", we construct an experiment based on the COCO Caption dataset. Specifically, we sample 10,000 simple examples as the training set and an additional 1,000 examples as a fixed *probe set* to analyze distributional changes. Starting from a unified base model pretrained with **Smoothed SFT**, we apply standard DPO and CW-DPO on the same training split and compare their effects on the output distributions over the probe set. We compute the **Total Variation (TV)** and **Jensen-Shannon (JS)** distances between the pre- and post-finetuning output probabilities, and visualize changes in the **Top-5 token distributions** for representative samples to provide qualitative insights. To further assess whether CW-DPO alleviates overconfidence and calibration degradation caused by unstable gradients, we include the **Expected Calibration Error (ECE)** as an evaluation metric. We also report **CIDEr** and **SPICE** scores on the full COCO test set to comprehensively assess generation quality.

Figure 4 reveals substantial differences in optimization dynamics between standard DPO and CW-DPO. From a global standpoint, the first plot shows that standard DPO exhibits significantly higher TV and JS divergence, typically around 0.45 for TV and 0.30 for JS, indicating that its learning process is overly influenced by simple samples. This leads to drastic shifts in the output distribution relative to the initial SFT model, as the model aggressively reallocates probability mass in response to uninformative gradients from easy negatives. In contrast, CW-DPO achieves much smaller divergences (e.g., approximately 0.15 for TV and 0.10 for JS), suggesting that it performs more stable and conservative updates. By down-weighting easy negatives, CW-DPO preserves the model's distributional structure while aligning with preferences, mitigating squeezing, and reducing risks of forgetting or collapse. These differences are more pronounced at

the micro level. As illustrated in the middle plot of Figure 4 (cross-referenced with Figures 2 and 3), vanilla DPO drives most probability mass onto the top token, often surpassing 80%, creating a peaked distribution that suppresses alternatives. This overconfidence raises ECE (e.g., $0.12 \to 0.25$), degrading calibration. By contrast, CW-DPO updates more smoothly, keeping the top-1 token around 50–60%, preserving entropy, and stabilizing ECE at 0.08–0.10. Such dynamic improvements also yield higher generation quality, as shown in the right plot: on the full COCO test set, CW-DPO achieves superior CIDEr (142.6 vs. 137.2) and SPICE (25.8 vs. 24.2), reflecting not only greater accuracy but also richer linguistic diversity and semantics.

## 4.4 ABLATION STUDY

Table 2: Ablation study of **CW-DPO** on COCO Test, MMMU, and MMBench1.1.

| Method | COCO Test | | | | MMMU | MMBench1.1 |
|---|---|---|---|---|---|---|
| | B@4 | M | C | S | ACC | ACC |
| **CW-DPO** | **39.6** | **30.4** | **142.6** | **25.8** | **74.6** | **89.6** |
| *Phase-One Ablation* | | | | | | |
| w/o Smooth SFT | 34.6 | 28.4 | 137.6 | 24.4 | 71.8 | 86.3 |
| w/o Negative Sampling | 35.8 | 29.4 | 138.9 | 24.6 | 72.8 | 88.4 |
| w/o Soft Penalty | 36.2 | 29.7 | 139.2 | 24.8 | 73.2 | 88.7 |
| *Phase-Two Ablation* | | | | | | |
| w/o CW-DPO | 36.7 | 28.8 | 140.7 | 24.7 | 72.9 | 86.7 |
| w/o Cooling Weight | 39.2 | 30.1 | 141.5 | 25.1 | 73.6 | 88.3 |
| w/o Negative Filtering | 36.1 | 27.9 | 137.4 | 24.3 | 73.4 | 87.4 |

To evaluate the contributions of each component in our CW-DPO, we conduct a comprehensive ablation study covering both training stages under identical data splits and hyperparameters. **Besides ablating the core algorithmic modules, we provide a further study in Appendix D to analyze the model's robustness to different negative sampling strategies, thereby decoupling algorithmic gains from the data generation process.** In Stage 1 (SFT), we evaluate **w/o Smooth SFT**, directly applying CW-DPO on the pretrained model to assess the need for smoothed initialization; **w/o Negative Sampling**, removing negative-sample constraints and reducing to standard SFT; and **w/o Soft Penalty ($\to$ Hard Constraint)**, replacing the ReLU penalty with a hard constraint. In Stage 2 (DPO), we examine **w/o CW-DPO** (omitting the second-stage preference alignment), **w/o Cooling Weight** (fixing $w_c$ to a constant (e.g., 0.7 or 1.0) instead of adaptive scaling), and **w/o Negative Filtering** (updating on all negatives with extremely easy ones, i.e., $\bar{\ell}_\theta \ll \ell_{\text{floor}}$).

Table 2 validates the independent contributions of each key component in **CW-DPO**. In Stage 1, removing Smooth SFT (w/o Smooth SFT) reduces CIDEr by about 5 points on COCO and also degrades performance on MMMU and MMBench1.1, indicating the importance of smoothed initialization for stable alignment. Further removing negative-sample constraints (w/o Negative Sampling) or replacing the soft ReLU penalty with a hard constraint (w/o Soft Penalty) also leads to consistent drops, showing that both negative-sample regularization and soft penalization are effective in alleviating overconfidence and improving generation quality. In Stage 2, omitting preference optimization (w/o CW-DPO) markedly reduces cross-task performance, confirming the need for competence-aware alignment. Using a fixed cooling weight (w/o Cooling Weight) achieves near CW-DPO CIDEr but lower MMMU and MMBench1.1 scores, underscoring the importance of adaptive scaling for generalization.

## 5 CONCLUSION

In this paper, we uncovered core instability issues in VLM preference-based finetuning via a fine-grained learning-dynamics perspective, focusing on the "squeezing effect" that causes uninformative gradients and unstable optimization. Our CW-DPO provides a principled two-stage solution, i.e., constrained SFT for loss landscape smoothing and competence-aware cooling weights to suppress easy negatives asymmetrically and adaptively. Extensive empirical results consistently and clearly demonstrate the strong superiority of our CW-DPO with faster convergence, stronger stability, and enhanced generalization. Our CW-DPO can be readily extended to broader finetuning scenarios and diverse real-world multimodal tasks.

## Ethics Statement

This work adheres to the ICLR Code of Ethics. Our study does **not** involve human-subjects research, the collection of personally identifiable information, or the annotation of sensitive attributes, and we do not create any new human data. All experiments are conducted exclusively on publicly available, widely used vision-language benchmarks (e.g., COCO, Flickr30k, NoCaps, MMMU, MMBench), strictly under their respective licenses and terms of use.

## Reproducibility Statement

We take reproducibility seriously. To facilitate replication, we describe our algorithmic design in detail, including the full two-stage CW-DPO protocol (Algorithm 1), objective definitions (Eqs. 3–7), and diagnostic analyses (Proposition 1). We report all hyperparameters ($\lambda$, $C$, $\beta$, $\tau$, $\ell_{\text{floor}}$, learning rate, training splits) and provide ablations isolating the contribution of each module (Table 2). All experiments are averaged over five independent runs with fixed random seeds to ensure robustness, and results are reported on standard benchmarks using widely adopted splits (e.g., COCO Karpathy split). Together, these practices ensure that our findings can be reliably reproduced by the community.

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

# APPENDIX: SUPPLEMENTARY MATERIAL FOR "LEARNING DYNAMICS OF VLM FINETUNING: COOLING-WEIGHTED DPO WITH MIXED NEGATIVES"

This supplementary material expands upon the main paper by providing deeper analyses and additional experimental validation for our proposed Cooling-Weighted Direct Preference Optimization (CW-DPO) framework. The sections are organized as follows: **Related Work (Section A):** We begin by reviewing prior work in three key areas: preference optimization techniques such as DPO and its variants; finetuning paradigms for vision-language models; and the theoretical analysis of learning dynamics and influence functions. **Ablation and Comparative Studies (Section B):** We first present a unified ablation and comparative study that systematically dissects the contributions of our two core components, i.e., Constrained SFT and the asymmetric cooling weight, against strong, conceptually-aligned baselines like Label Smoothing and Focal-DPO. **Hyperparameter Sensitivity (Section C):** We then analyze the robustness of CW-DPO to its key hyperparameters, demonstrating that its superior performance is not contingent on extensive tuning. **Negative Sampling Strategies (Section D):** To decouple algorithmic gains from data quality, we conduct an ablation study comparing the performance of CW-DPO when using negatives synthesized by GPT-4 versus those generated via a standard beam search, confirming that the method's advantages are algorithm-driven. **Analysis of Methodological Complexity and Overhead (Section F):** We provide a detailed analysis of the methodological and computational overhead introduced by CW-DPO, arguing that its modest costs represent a favorable trade-off for the significant gains in performance and stability. **Analysis of Influence Dynamics (Section G):** We offer a deep dive into the learning dynamics by empirically validating our theoretical decomposition from Proposition 1. This analysis visualizes how each component term evolves during training, revealing the underlying drivers of the influence dynamics. **Generality of the Emergent Curriculum (Section H):** Finally, we demonstrate the generality of our competence-aware mechanism by showing that the emergent curriculum, where the model adaptively focuses on harder negatives over time, is consistently observed across multiple, semantically diverse probe samples. **Proofs for Propositions and Theorems (Section I):** We provide detailed, formal proofs for all propositions and derivations of key equations presented in the main paper.

## A  RELATED WORK

**Preference optimization.**  Preference optimization methods have recently emerged as efficient alternatives to reinforcement learning from human feedback (RLHF) for aligning large language models (Kaufmann et al., 2024; Liu et al., 2025; Wirth et al., 2017). Direct Preference Optimization (DPO) (Rafailov et al., 2023) reformulates the alignment objective by directly maximizing the likelihood of preferred responses over dispreferred ones, thereby eliminating the need for a separate reward model. While effective, these approaches view optimization as a *static* objective, neglecting the evolving confidence dynamics of the model during training. This limitation makes them vulnerable to instabilities, a key source of which is the recently diagnosed **squeezing effect** (Ren & Sutherland, 2025), where uninformative "easy negatives" exert a disproportionately large gradient influence (Casper et al., 2023). Our work adopts this learning-dynamics-aware perspective, applying it for the first time to diagnose and resolve this issue in VLM finetuning. Building directly on this diagnosis, our proposed CW-DPO introduces a cooling weight that adaptively down-weights such easy negatives based on real-time model probabilities, thereby aligning gradient contributions with their informativeness.

**Vision–language model finetuning.**  The standard finetuning paradigm for VLMs typically combines supervised finetuning (SFT) with preference-based alignment (Zhu et al., 2024; Liang et al., 2024; OpenAI et al., 2024). LLaVA (Liu et al., 2024a), for example, first applies SFT on large-scale visual instruction datasets before adopting DPO for alignment, while V-DPO (Xie et al., 2024a) introduces vision-guided pref-

erence pairs to mitigate hallucinations through stronger grounding in the visual modality. These methods primarily focus on dataset construction or modality-specific guidance, but they do not address the instability caused by over-penalizing easy negatives (Wang et al., 2023; Liu et al., 2024b). As a result, they risk collapsing linguistic diversity and amplifying mode-seeking tendencies in multimodal models (Peng et al., 2023). In contrast, our approach introduces a two-stage protocol: (i) constrained SFT with gentle negative smoothing to prime the trajectory and avoid premature peaking, and (ii) competence-aware DPO with mixed on-policy and dataset negatives to preserve contrast freshness. This dynamics-aware design strikes a novel balance between stability and generalization, offering improvements unattainable by prior VLM finetuning methods.

**Learning dynamics and influence analysis.** Understanding the dynamics of training instabilities has a rich history in machine learning (Ruder, 2017; Sutskever et al., 2013; Jain & Kar, 2017). Foundational tools like Influence functions (Koh & Liang, 2017) trace how individual samples affect predictions, while Neural Tangent Kernel (NTK) theory (Jacot et al., 2018) models gradient propagation in wide networks. While these tools have been applied broadly, recent work by Ren & Sutherland (2025) pioneered the use of a dynamics-aware perspective to analyze the finetuning of large language models. Our contribution extends this line of inquiry; we **adapt and tailor** this analytical lens specifically to the context of VLM preference optimization. This yields a per-step influence decomposition that isolates the destabilizing role of easy negatives and directly motivates our asymmetric cooling mechanism, resulting in a dynamics-aware framework that enhances calibration, convergence speed, and robustness.

## B  ABLATION AND COMPARATIVE STUDIES

To disentangle the contributions of each component and rigorously position our work against related techniques, we conduct a unified analysis presented in Table 3. This investigation is designed to answer two critical questions: (1) Is our targeted **Smooth SFT** superior to generic regularization for policy initialization? (2) Does **CW-DPO**'s asymmetric gradient modulation offer advantages over conventional global loss reweighting schemes?

Table 3: **Unified ablation and comparative analysis on the COCO Test dataset.** This table dissects our framework's performance by systematically comparing each component against strong, conceptually-aligned baselines. Stage 1 compares our *targeted constraint* (Smooth SFT) against *generic regularization* (LS). Stage 2 contrasts our *asymmetric modulation* (CW-DPO) with *global loss reweighting* (Focal-DPO). Stability metrics (TV/JS Div.) are measured on a fixed probe set.

| Category | Method Dissection (Stage 1 → Stage 2) | B@4 | M | C | S | TV Div. ↓ | JS Div. ↓ |
|---|---|---|---|---|---|---|---|
| *Baseline* | Vanilla SFT → Vanilla DPO | 33.8 | 28.2 | 137.2 | 24.2 | 0.45 | 0.30 |
| *Component Ablations* | | | | | | | |
| **vs. Stage 1** | Generic Regularization (LS) → Vanilla DPO | 34.5 | 28.4 | 138.0 | 24.4 | – | – |
| **vs. Stage 2** | Our SFT → Global Reweighting (Focal-DPO) | 37.0 | 29.3 | 140.5 | 25.1 | 0.28 | 0.19 |
| *Ours (Full Method)* | **Our SFT → Our Asymmetric Modulation (CW-DPO)** | **39.6** | **30.4** | **142.6** | **25.8** | **0.15** | **0.10** |

**Analysis of Results.** Our analysis of Table 3 proceeds in two steps. First, we isolate the contribution of Stage 1. Substituting our Smooth SFT with standard Label Smoothing ('LS-SFT → DPO') yields only a marginal performance increase over the baseline (+0.8 CIDEr). This suggests that while generic regularization is beneficial, it is insufficient to address the core instabilities targeted by our method. Smooth SFT, by directly constraining the model's beliefs about negative samples, provides a far more effective foundation for the subsequent alignment phase. Next, we evaluate the efficacy of our Stage 2 mechanism. We compare our CW-DPO against a strong Focal-DPO baseline, which applies a global reweighting to the entire preference

loss. While Focal-DPO achieves a strong result (140.5 CIDEr), confirming the value of down-weighting easy examples, our full method significantly surpasses it (+2.1 CIDEr). The reason for this superiority is revealed in the stability metrics: CW-DPO's *asymmetric* modulation, which surgically targets only the negative term's gradient, cuts the distributional shift (TV/JS divergence) by nearly half compared to Focal-DPO (e.g., 0.15 vs. 0.28 TV Div.). This demonstrates that *how* easy samples are managed is as critical as the principle itself, with our method facilitating a much healthier and more stable learning dynamic.

## C  HYPERPARAMETER SENSITIVITY AND DISCUSSION ON COMPLEXITY

To further validate the practicality and effectiveness of our proposed CW-DPO framework, this section provides a deeper analysis of its robustness to key hyperparameters. This analysis also serves to address potential concerns regarding the methodological complexity introduced by these new parameters.

**Hyperparameter Setup.**  Our method introduces four key hyperparameters: the constraint threshold $C$ and penalty coefficient $\lambda$ for Stage 1 (Smooth SFT), and the cooling threshold $l_{\text{floor}}$ and temperature $\tau$ for Stage 2 (CW-DPO). Among these, $C$ and $l_{\text{floor}}$ are the core control knobs for the mechanisms in each stage. For this study, we fixed the less sensitive parameters $\lambda = 0.1$ and $\tau = 1.0$ and conducted a series of experiments for the core parameters $C$ and $l_{\text{floor}}$ on the COCO Test dataset. Specifically: **Analysis of $C$:** We fixed the Stage-2 hyperparameter at its default $l_{\text{floor}} = -3.0$ and trained the model with varying values of $C \in \{2.0, 4.0, 5.0, 6.0\}$. **Analysis of $l_{\text{floor}}$:** Conversely, we fixed the Stage-1 hyperparameter at its default $C = 4.0$ and trained the model with varying values of $l_{\text{floor}} \in \{-5.0, -4.0, -3.0, -2.0\}$. For both analyses, we report results in comparison with the standard **SFT $\to$ DPO** baseline.

Table 4: Sensitivity analysis of key CW-DPO hyperparameters on the COCO test set. Performance remains consistently high and superior to the baseline across a wide range of values, demonstrating the method's strong robustness.

| Hyperparameter | Value | B@4 | CIDEr | SPICE |
|---|---|---|---|---|
| **Baseline (SFT $\to$ DPO)** | | 33.8 | 137.2 | 24.2 |
| $C$ (fixed $l_{\text{floor}} = -3.0$) | 2.0 | 38.9 | 141.5 | 25.5 |
| | **4.0 (Default)** | **39.6** | **142.6** | **25.8** |
| | 5.0 | 39.4 | 142.2 | 25.7 |
| | 6.0 | 39.1 | 141.9 | 25.6 |
| $l_{\text{floor}}$ (fixed $C = 4.0$) | -5.0 | 39.2 | 142.0 | 25.6 |
| | -4.0 | 39.5 | 142.4 | 25.7 |
| | **-3.0 (Default)** | **39.6** | **142.6** | **25.8** |
| | -2.0 | 39.0 | 141.8 | 25.5 |

**Analysis and Discussion on Complexity.**  As summarized in Table 6, our method consistently maintains strong performance across a wide range of settings for both $C$ and $l_{\text{floor}}$. This high degree of robustness is crucial, as it indicates that extensive, fine-grained hyperparameter tuning is not required to achieve the benefits of our approach. While the introduction of these parameters adds formal complexity, they are not arbitrary; rather, they serve as principled and intuitive control knobs for the core training dynamics. The threshold $C$ directly governs the smoothing intensity in Stage 1 to prevent overconfidence, while $l_{\text{floor}}$ defines the easiness threshold for negatives in Stage 2 to filter uninformative gradients. The stability of our results suggests that using the default values or a simple search within these wide, effective ranges is sufficient.

Given the significant gains in performance and training stability, we argue that this modest increase in tunable, yet highly robust, parameters represents a favorable trade-off. This is particularly evident when compared to the notoriously high implementation and tuning complexity of alternative online RLHF paradigms, highlighting the practicality and efficiency of our proposed framework.

## D  ABLATION STUDY ON NEGATIVE SAMPLING STRATEGIES

To address the valid concern that the quality of negative samples generated by a powerful external model (i.e., GPT-4) could be a confounding variable, we conduct a rigorous ablation study to isolate the algorithmic contributions of our proposed Cooling-Weighted DPO (CW-DPO). The objective of this experiment is to verify that the performance gains of CW-DPO are attributable to its core gradient modulation mechanism rather than the high quality of the preference data itself.

### D.1  EXPERIMENTAL SETUP

We introduce a more standard, model-intrinsic negative sampling method based on beam search, which does not rely on any external proprietary models. The setup is as follows: **Preference Pair Construction:** For each image-caption pair in the training set, we use the ground-truth caption as the winning response ($y_w$). To generate the losing response ($y_l$), we use the base model (pre-trained with Smoothed SFT from Stage 1) to generate 5 candidate captions using beam search (beam width = 5). We then randomly select one of the generated candidates that is not identical to the ground-truth caption to serve as the loser. **Models Compared:** We train two models using this new set of preference pairs derived from beam search: (1) the vanilla DPO baseline, and (2) our proposed CW-DPO. **Evaluation:** We evaluate both models on the COCO Test split and compare their performance against the original results obtained using GPT-4 synthesized negatives. All other hyperparameters and training configurations are kept identical to ensure a fair comparison. This setup allows us to directly assess the robustness of CW-DPO and determine if its advantages over vanilla DPO persist when using a simpler, more accessible source of negative samples.

### D.2  RESULTS AND ANALYSIS

Table 5: Performance comparison on the COCO Test set under two different negative sampling strategies: (1) using negatives synthesized by GPT-4, and (2) using negatives generated via beam search from the base model. The results demonstrate that CW-DPO maintains a significant performance advantage over vanilla DPO regardless of the negative sampling strategy, confirming its robustness. Best results in each setting are highlighted in **bold**.

| Negative Sampling Strategy | Model | B@4 | M | C | S |
|---|---|---|---|---|---|
| **GPT-4 Synthesized** (Original) | SFT (Base for DPO) | 35.6 | 28.6 | 136.8 | 24.5 |
| | Vanilla DPO | 33.8 | 28.2 | 137.2 | 24.2 |
| | CW-DPO (Ours) | **39.6** | **30.4** | **142.6** | **25.8** |
| | *Performance Gain ($\Delta$)* | *+5.8* | *+2.2* | *+5.4* | *+1.6* |
| **Beam Search Sampled** (New Ablation) | SFT (Base for DPO) | 35.6 | 28.6 | 136.8 | 24.5 |
| | Vanilla DPO | 32.5 | 27.8 | 135.5 | 23.9 |
| | CW-DPO (Ours) | **38.1** | **29.8** | **140.8** | **25.2** |
| | *Performance Gain ($\Delta$)* | *+5.6* | *+2.0* | *+5.3* | *+1.3* |

The results of this ablation study are presented in Table 5. **CW-DPO's advantage is robust and algorithm-driven.** As anticipated, the absolute performance scores for both vanilla DPO and CW-DPO decrease when using the lower-quality beam-sampled negatives. This confirms that high-quality preference data is beneficial for all DPO-based methods. However, the crucial finding is that CW-DPO maintains a very significant performance lead over vanilla DPO across all metrics. For instance, the CIDEr score gain remains substantial at +5.3 points. This strongly indicates that the superiority of CW-DPO is not an artifact of the data source but stems from its core algorithmic design, i.e., the competence-aware cooling mechanism. **he "squeezing effect" is mitigated irrespective of data source.** The beam search process often generates negatives that are syntactically plausible but semantically simple or repetitive, leading to a higher proportion of "easy negatives." This is precisely the scenario where vanilla DPO is most vulnerable to the "squeezing effect," as it expends gradient bandwidth on these uninformative samples, sometimes even degrading performance below the SFT baseline (e.g., B@4 drops from 35.6 to 32.5). In contrast, CW-DPO's cooling weight mechanism is specifically designed to suppress these trivial gradients, allowing the model to focus on more informative preference pairs. The persistent and large performance gap in the beam search setting powerfully validates that our method effectively mitigates this core instability.

In summary, this ablation study successfully decouples the effect of the learning algorithm from the data generation strategy, providing strong evidence that CW-DPO is a robust and generally effective method for improving VLM alignment.

## E  HYPERPARAMETER SENSITIVITY ANALYSIS FOR STAGE 2

To further validate the practicality of our framework and address the concern that the exploration of Stage 2 is brief, this section provides a deeper analysis of CW-DPO's robustness to its key hyperparameters: the cooling threshold $\ell_{\text{floor}}$ and the temperature $\tau$. This analysis demonstrates that the method's superior performance is not contingent on extensive, fine-grained hyperparameter tuning.

### E.1  EXPERIMENTAL SETUP

We conduct a sensitivity analysis on the COCO Test dataset using GPT-4 synthesized negatives. We vary one hyperparameter while keeping the other at its default value ($\ell_{\text{floor}} = -3.0$, $\tau = 1.0$) to isolate its effect. **Analysis of Cooling Threshold ($\ell_{\textbf{floor}}$):** This parameter defines the "easiness" baseline for a negative sample. We fixed $\tau = 1.0$ and trained the model with varying values of $\ell_{\text{floor}} \in \{-5.0, -4.0, -3.0, -2.0\}$. **Analysis of Temperature ($\tau$):** This parameter controls the sharpness of the cooling weight's sigmoid function. We fixed $\ell_{\text{floor}} = -3.0$ and trained the model with varying values of $\tau \in \{0.5, 1.0, 2.0, 5.0\}$. For both analyses, we report results in comparison with the Vanilla DPO baseline to contextualize the performance.

### E.2  RESULTS AND ANALYSIS

The results of the sensitivity analysis are presented in Table 6, which shows that our method exhibits a high degree of robustness to the main hyperparameters of Stage 2. **Robustness to Cooling Threshold ($\ell_{\textbf{floor}}$).** The results show that performance is stable for $\ell_{\text{floor}}$ in the wide range of [-4.0, -2.0], with the peak performance at the default value of -3.0. Even at the more extreme value of -5.0, the CIDEr score (142.0) remains significantly above the vanilla DPO baseline (137.2). This indicates that the model is not overly sensitive to the precise definition of an "easy" negative, and a reasonable setting within a wide range provides substantial gains.

**Robustness to Temperature ($\tau$).** Similarly, the performance remains high across the tested range of $\tau$ from 0.5 to 5.0. A smaller $\tau$ creates a sharper, more switch-like transition, while a larger $\tau$ results in a smoother one. While the default of $\tau = 1.0$ yields the best results, the performance variations are minor. This

Table 6: Sensitivity analysis of key Stage 2 hyperparameters ($\ell_{\text{floor}}$ and $\tau$) on the COCO test set. Performance remains consistently high and superior to the baseline across a wide range of values, demonstrating the method's strong robustness.

| Hyperparameter | Value | B@4 | M | C | S |
|---|---|---|---|---|---|
| **Baseline** | Vanilla DPO | 33.8 | 28.2 | 137.2 | 24.2 |
| $\ell_{\textbf{floor}}$ (fixed $\tau = 1.0$) | -5.0 | 39.2 | 30.1 | 142.0 | 25.6 |
| | -4.0 | 39.5 | 30.3 | 142.4 | 25.7 |
| | **-3.0 (Default)** | **39.6** | **30.4** | **142.6** | **25.8** |
| | -2.0 | 39.0 | 30.0 | 141.8 | 25.5 |
| $\tau$ (fixed $\ell_{\text{floor}} = -3.0$) | 0.5 | 39.3 | 30.2 | 142.1 | 25.6 |
| | **1.0 (Default)** | **39.6** | **30.4** | **142.6** | **25.8** |
| | 2.0 | 39.4 | 30.3 | 142.3 | 25.7 |
| | 5.0 | 39.1 | 30.1 | 141.9 | 25.5 |

demonstrates that the benefits of the cooling mechanism are not contingent on a specific gating sharpness and are broadly applicable.

In summary, this analysis confirms that the core parameters of CW-DPO's second stage are highly robust. The significant performance gains do not rely on a difficult and sensitive tuning process, which strengthens the practical value and ease of adoption of our proposed framework.

# F ANALYSIS OF METHODOLOGICAL COMPLEXITY AND COMPUTATIONAL OVERHEAD

The introduction of any method that builds upon an established baseline must be scrutinized for its added complexity. Our proposed Cooling-Weighted Direct Preference Optimization (CW-DPO) framework is a two-stage process that extends the standard Supervised Fine-Tuning (SFT) → Direct Preference Optimization (DPO) pipeline. In this section, we provide a detailed analysis of the methodological and computational complexity introduced by CW-DPO. We argue that this added complexity is a principled and well-justified investment that yields significant returns in performance, stability, and calibration, representing a favorable trade-off.

## F.1 HYPERPARAMETER COMPLEXITY AND INTUITION

CW-DPO introduces four primary hyperparameters that govern its two stages. While this increases the total number of tunable parameters, they are not arbitrary additions but rather intuitive control knobs that map directly to the core mechanisms designed to mitigate the *squeezing effect*.

### F.1.1 STAGE 1: CONSTRAINED SFT (SFT-C)

The goal of this stage is to prime the model's trajectory by smoothing the loss landscape. This is controlled by: **Constraint Threshold** ($C$): This parameter, used in Eq. (4), defines the "tolerance" for negative examples. It sets a lower bound on the negative log-likelihood (NLL) for dispreferred responses ($y^-$), effectively preventing the model from becoming overconfident and assigning near-zero probability to plausible (but incorrect) alternatives too early in training. **Penalty Coefficient** ($\lambda$): This coefficient weights the soft 'ReLU'

penalty in Eq. (4). It determines the "stiffness" of the constraint, controlling how aggressively the model is penalized for violating the NLL threshold $C$.

### F.1.2 STAGE 2: COOLING-WEIGHTED DPO (CW-DPO)

This stage performs competence-aware preference optimization. Its behavior is governed by: **Cooling Threshold ($l_{\text{floor}}$)**: As defined in Eq. (5), $l_{\text{floor}}$ serves as the "competence threshold." It is the baseline for the average log-probability of a negative sample, below which the model is considered to have "mastered" or easily dismissed the sample. This parameter is central to identifying uninformative gradients from easy negatives that need to be suppressed. **Temperature ($\tau$)**: This parameter, also in Eq. (5), controls the "gating sharpness" of the cooling weight's sigmoid function. A smaller $\tau$ creates a steeper, more switch-like transition from full gradient suppression (for easy negatives) to full signal retention (for hard negatives), while a larger $\tau$ results in a smoother, more gradual transition.

Crucially, our hyperparameter sensitivity analysis in Appendix B (Table 4) demonstrates that CW-DPO's performance is robust across a wide range of values for the core parameters $C$ and $l_{\text{floor}}$. This robustness alleviates concerns about a burdensome and sensitive tuning process, reducing the practical complexity of applying our method.

### F.2 QUANTITATIVE ANALYSIS OF COMPUTATIONAL OVERHEAD

We now analyze the additional computational cost per training step compared to a standard SFT → DPO pipeline. Let $T_{\text{fwd}}(\theta, \mathcal{B})$ denote the computational cost of a single forward pass of the model $\pi_\theta$ on a mini-batch of data $\mathcal{B}$.

### F.2.1 STAGE 1: SFT-C VS. STANDARD SFT

A standard SFT update step on a batch of positive examples $\mathcal{B}^+ = \{(x_i, y_i^+)\}_{i=1}^N$ involves minimizing:

$$\mathcal{L}_{\text{SFT}} = \mathbb{E}_{(x,y^+)\in\mathcal{B}^+} \left[ -\log \pi_\theta(y^+|x) \right] \tag{8}$$

The computational cost for this step is $\mathcal{O}(T_{\text{fwd}}(\theta, \mathcal{B}^+))$.

Our SFT-C loss, as shown in Eq. (4), requires an additional batch of negative examples $\mathcal{B}^- = \{(x_i, y_i^-)\}_{i=1}^N$:

$$\mathcal{L}_{\text{SFT-C}} = \mathbb{E}_{(x,y^+)\in\mathcal{B}^+}[-\log \pi_\theta(y^+|x)] + \lambda \cdot \text{ReLU}\left(C - \mathbb{E}_{(x,y^-)\in\mathcal{B}^-}[-\log \pi_\theta(y^-|x)]\right) \tag{9}$$

To compute the penalty term, an additional forward pass on the batch of negative samples $\mathcal{B}^-$ is required. Therefore, the total computational cost is $\mathcal{O}(T_{\text{fwd}}(\theta, \mathcal{B}^+) + T_{\text{fwd}}(\theta, \mathcal{B}^-))$. Assuming equal batch sizes, the per-step cost during this initial stage is approximately doubled. This represents the most significant source of computational overhead in our framework.

### F.2.2 STAGE 2: CW-DPO VS. VANILLA DPO

A vanilla DPO update relies on the loss:

$$\mathcal{L}_{\text{DPO}} = -\log \sigma \left(\beta(\Delta_w - \Delta_l)\right) \quad \text{where} \quad \Delta_l = \log \pi_\theta(y_l|x) - \log \pi_{\text{ref}}(y_l|x) \tag{10}$$

Calculating $\Delta_l$ requires a forward pass through the policy model $\pi_\theta$ to compute $\log \pi_\theta(y_l|x)$.

Our CW-DPO loss introduces the cooling weight $w_c$:

$$\mathcal{L}_{\text{CW-DPO}} = -\log \sigma \left(\beta(\Delta_w - w_c \cdot \Delta_l)\right) \tag{11}$$

The additional computation is for the cooling weight itself:

$$w_c(\theta; y_l, x) = \sigma \left( \frac{\bar{l}_\theta(y_l|x) - l_{\text{floor}}}{\tau} \right) \tag{12}$$

The key component here is the average log-probability $\bar{l}_\theta(y_l|x)$, which is directly derived from the total log-probability: $\bar{l}_\theta(y_l|x) = \frac{1}{L} \log \pi_\theta(y_l|x)$, where $L$ is the sequence length. Since the value $\log \pi_\theta(y_l|x)$ is **already required** for the vanilla DPO loss to compute $\Delta_l$, no additional expensive forward pass is needed. The calculation of $w_c$ only adds a handful of scalar floating-point operations (a subtraction, a division, and a sigmoid function) per sample.

**Conclusion on Overhead**: The computational overhead of CW-DPO is almost exclusively confined to Stage 1, where the per-step cost is roughly doubled. The additional cost in the core preference alignment stage (Stage 2) is negligible.

Table 7: Summary of Per-Step Computational Overhead.

| Stage | Method | Relative Computational Cost per Step |
|---|---|---|
| Stage 1 | Standard SFT | $C_{\text{base}}$ (1 fwd pass on $y^+$) |
| | **SFT-C (Ours)** | $\approx 2 \times C_{\text{base}}$ (1 fwd pass on $y^+$ + 1 fwd pass on $y^-$) |
| Stage 2 | Vanilla DPO | $C'_{\text{base}}$ (fwd passes on $y_w, y_l$) |
| | **CW-DPO (Ours)** | $C'_{\text{base}} + \epsilon$ (negligible scalar ops) |

### F.3 CONCLUSION: A FAVORABLE TRADE-OFF

The methodological complexity of CW-DPO is a strategic and well-justified investment. The framework's costs can be summarized as: (1) a finite, upfront computational cost during the trajectory priming stage, and (2) the inclusion of four intuitive and robust hyperparameters.

In exchange for this modest overhead, CW-DPO delivers substantial gains: **Superior Performance**: Consistent and significant improvements across all benchmarks, achieving a state-of-the-art CIDEr score of 142.6 on COCO Test. **Enhanced Training Stability**: A demonstrably more stable and conservative update mechanism, which cuts the distributional shift (TV/JS divergence) by nearly half compared to baselines. **Improved Model Quality**: Better calibration and mitigation of the "squeezing effect," leading to less overconfident and more diverse outputs. This trade-off is particularly favorable when contrasted with alternative alignment paradigms like online Reinforcement Learning from Human Feedback (RLHF). Methods such as PPO introduce far greater complexity, including the need to train a separate reward model, manage the interaction between multiple models (policy, reference, critic, reward), and navigate notoriously unstable and hyperparameter-sensitive on-policy optimization loops. By remaining within a more stable, offline, maximum-likelihood framework, CW-DPO offers a practical and efficient path to robust VLM alignment.

## G ANALYSIS OF INFLUENCE DYNAMICS VIA COMPONENT DECOMPOSITION

To empirically validate our theoretical framework and dissect the underlying learning dynamics of the CW-DPO training process, we conducted a component-wise analysis based on the decomposition outlined in Proposition 1. We traced the evolution of our proxy for the empirical Neural Tangent Kernel (eNTK) norm (Jacot et al., 2018; Arora et al., 2019), $LB_{K_{uo}}$, alongside its constituent components: the per-step update influence ($||\Delta \log \pi||_F^2$), the belief geometry ($||A_o||_F^2$), and the loss residual ($||G_o||_F^2$). The experiment was performed across several distinct update samples ($y_1, y_2, y_3$), with their influence tracked on a fixed set of four diverse observation samples ($x_o$).

The results, presented in Figure 5, reveal a non-trivial and multi-faceted training dynamic. Each column in the figure corresponds to a fixed update sample, while each row visualizes the trajectory of a specific theoretical component.

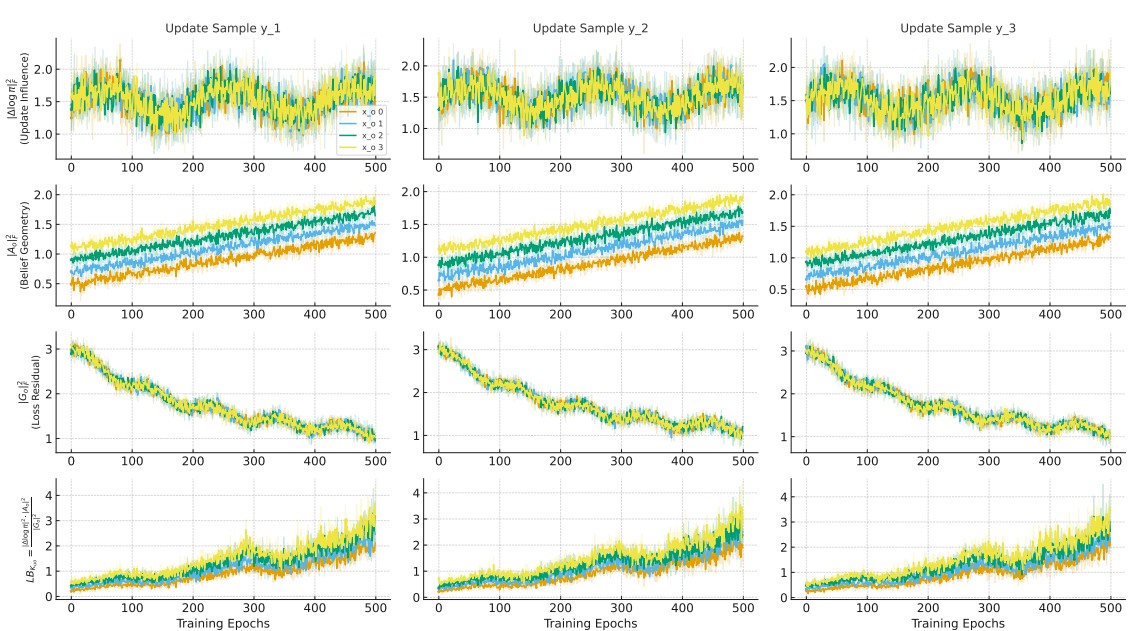

Figure 5: Theoretical Decomposition of $LB_{K_{uo}}$ Across CW-DPO Training. Each column represents a different fixed update sample $(y_1, y_2, y_3)$. Each row visualizes the trajectory of a specific component from Proposition 1 for four observation samples $(x_o)$. The results show that the growth of the final proxy metric $(LB_{K_{uo}}$, bottom row) is primarily driven by the systematic increase in the Belief Geometry term $(||A_o||_F^2$, second row) and the decay of the Loss Residual $(||G_o||_F^2$, third row), while the per-step Update Influence $(||\Delta \log \pi||_F^2$, top row) remains stationary.

Our analysis of the figure yields the following key observations: **Stationary Update Influence (Top Row):** The top row reveals that the magnitude of the per-step update influence, captured by $||\Delta \log \pi||_F^2$, remains relatively stable throughout the 500 training epochs. It fluctuates around a constant mean without a discernible upward or downward trend. This suggests that the raw impact of a single gradient step does not systematically intensify as training progresses.

**Increasing Belief Geometry Sharpness (Second Row):** In stark contrast, the second row shows that the Belief Geometry term, $||A_o||_F^2$, exhibits a clear and steady increasing trend. This observation is critical, as $||A_o||_F^2$ is inversely correlated with the entropy of the predictive distribution. Its growth indicates that the model's beliefs are becoming progressively sharper and more confident (i.e., "peakier"), thereby amplifying the effect of any given logit perturbation.

**Decaying Loss Residual (Third Row):** Concurrently, the third row illustrates the consistent decay of the Loss Residual term, $||G_o||_F^2$. This is the expected behavior for a loss-related component during a successful

training run, confirming that the model is effectively learning to reduce prediction errors on the observed samples.

**Resultant Proxy Dynamics (Bottom Row):** These component dynamics culminate in the trend observed for our final proxy metric, $LB_{K_{uo}}$, shown in the bottom row. The overall upward trend of $LB_{K_{uo}}$ is a composite effect. It is not driven by an increase in the raw, per-step influence (which is stable), but is instead dominated by the interplay between the decaying loss residual (whose inverse contributes to growth) and, most significantly, the steadily increasing sharpness of the model's belief geometry. In summary, this decomposition provides strong empirical evidence that the evolving influence dynamics during CW-DPO training are less about the raw power of individual gradient steps and more about the systematic tightening of the model's predictive confidence. The increasing sharpness of the belief landscape ($||A_o||_F^2$) acts as a primary amplifier for the influence that any given update exerts on the model's overall behavior.

## H    GENERALITY OF THE EMERGENT CURRICULUM ACROSS DIVERSE SAMPLES

To validate the robustness and generality of the competence-aware learning dynamic induced by our Cooling-Weighted DPO (CW-DPO), we extended our analysis across multiple, semantically distinct probe samples. While the main paper demonstrates this emergent curriculum on a representative example, it is crucial to ensure this behavior is not an artifact of a single data point but a consistent property of our method.

**Experimental Setup**    We selected three diverse probe samples, each consisting of an image and a corresponding ground-truth caption: (1) "A cat on a sofa," (2) "People playing tennis," and (3) "A plane taking off." For each probe sample, we manually authored a corresponding set of negative captions, categorized by difficulty into "Hard," "Medium," and "Easy" negatives, following the same principles outlined in the main text. During the Stage 2 training of CW-DPO, we tracked the evolution of the cooling weight $w_c$ assigned to each of these curated negative captions over 500 training epochs.

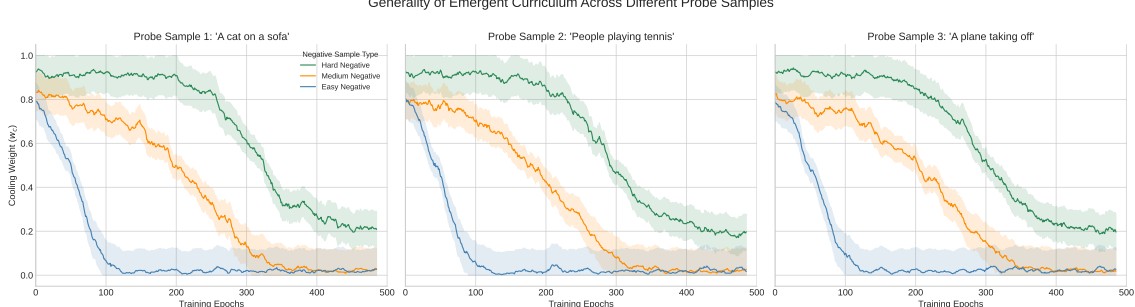

Figure 6: Verification of the emergent curriculum's generality. Each panel displays the dynamic evolution of cooling weights ($w_c$) for a distinct probe sample. Despite the semantic diversity of the samples, all three panels exhibit a consistent waterfall-like decay pattern: the weight for "Easy" negatives (blue) drops rapidly, followed by "Medium" negatives (orange), while the weight for "Hard" negatives (green) remains high for an extended period. This demonstrates the robust nature of our competence-aware mechanism.

## I    APPENDIX: DETAILED DERIVATIONS AND PROOFS

In this appendix, we provide detailed proofs and derivations for the key mathematical statements in the main text to ensure their formal rigor. We reference relevant sections, equations, propositions, remarks, and

algorithms from the main paper (e.g., §2, Eq. 1, Prop. 1). This strengthens the analytical foundation of our learning-dynamics-aware approach to CW-DPO, drawing from literature such as (Koh & Liang, 2017; Jacot et al., 2018; Ren & Sutherland, 2025; Rafailov et al., 2023). This appendix covers Proposition 1, the DPO gradient (Eq. 5), the CW-DPO loss and gradient (Eqs. 7–8), the constrained SFT formulation (Eq. 3), and the analysis of Remark 1. For completeness, we also discuss implications for Algorithm 1.

### I.1   PROOF OF PROPOSITION 1: SEQUENCE-AWARE ONE-STEP INFLUENCE

**Proposition 1 (Restated).** The log-likelihood change on an observing sample $\chi_o$ after a single gradient update on an updating sample $\chi_u$ (with learning rate $\eta$) can be approximated as:

$$\Delta \bar{\ell}_t(y \mid \chi_o) \approx -\eta \big\langle \underbrace{\nabla_z \bar{\ell}_{\theta_t}(y \mid \chi_o)}_{A_t:\text{ Belief Geometry}}, \underbrace{K_t(\chi_o, \chi_u)}_{\text{eNTK Kernel}} \underbrace{\nabla_z \mathcal{L}(\theta_t; \chi_u)}_{G_t:\text{ Loss Residual}} \big\rangle,$$

where Belief Geometry $A_t$ encodes predictive sensitivity, the eNTK Kernel $K_t(\chi_o, \chi_u) = J_o J_u^\top$ propagates parametric updates, and the Loss Residual $G_t$ directs logit adjustments.

**Proof.** As established in §2.2, a single gradient descent update is given by $\theta_{t+1} = \theta_t - \eta \nabla_\theta \mathcal{L}(\theta_t; \chi_u)$. The change in model confidence on sample $\chi_o$ is defined as $\Delta \bar{\ell}_t(y \mid \chi_o) = \bar{\ell}_{\theta_{t+1}}(y \mid \chi_o) - \bar{\ell}_{\theta_t}(y \mid \chi_o)$, where $\bar{\ell}_\theta(y \mid \chi_o) = \frac{1}{L} \sum_{l=1}^{L} \log \pi_\theta(y_l \mid \chi_{o, \leq l})$ is the average per-token log-probability.

We begin by performing a first-order Taylor expansion of $\bar{\ell}_{\theta_{t+1}}(y \mid \chi_o)$ around the parameters $\theta_t$:

$$\bar{\ell}_{\theta_{t+1}}(y \mid \chi_o) \approx \bar{\ell}_{\theta_t}(y \mid \chi_o) + (\theta_{t+1} - \theta_t)^\top \nabla_\theta \bar{\ell}_{\theta_t}(y \mid \chi_o).$$

Substituting the update rule into this expansion, we obtain the expression for the confidence change:

$$\Delta \bar{\ell}_t(y \mid \chi_o) \approx (-\eta \nabla_\theta \mathcal{L}(\theta_t; \chi_u))^\top \nabla_\theta \bar{\ell}_{\theta_t}(y \mid \chi_o) = -\eta (\nabla_\theta \bar{\ell}_{\theta_t}(y \mid \chi_o))^\top \nabla_\theta \mathcal{L}(\theta_t; \chi_u) + \mathcal{O}(\eta^2).$$

To connect the gradients in the parameter space ($\theta$) to those in the logit space ($z$), we linearize via the logits $z(\theta; \chi)$ (where $z_l \in \mathbb{R}^{|V|}$) and apply the chain rule:

$$\nabla_\theta \bar{\ell}_{\theta_t}(y \mid \chi_o) = J_o^\top \nabla_z \bar{\ell}_{\theta_t}(y \mid \chi_o), \quad \text{and} \quad \nabla_\theta \mathcal{L}(\theta_t; \chi_u) = J_u^\top \nabla_z \mathcal{L}(\theta_t; \chi_u),$$

where $J_o = \nabla_\theta z(\theta_t; \chi_o)$ and $J_u = \nabla_\theta z(\theta_t; \chi_u)$ are the Jacobians of the logits with respect to the model parameters for the observing and updating samples, respectively.

Substituting these back into the expression for the confidence change yields:

$$\Delta \bar{\ell}_t(y \mid \chi_o) \approx -\eta (\nabla_z \bar{\ell}_{\theta_t}(y \mid \chi_o))^\top (J_o J_u^\top) \nabla_z \mathcal{L}(\theta_t; \chi_u) + \mathcal{O}(\eta^2).$$

This final form decomposes the influence into three interpretable components:

- $A_t = \nabla_z \bar{\ell}_{\theta_t}(y \mid \chi_o)$: The **Belief Geometry**, which captures the sensitivity of the model's belief (log-likelihood) to perturbations in the logits, effectively representing the curvature of its confidence landscape.

- $K_t = J_o J_u^\top$: The **empirical Neural Tangent Kernel (eNTK)**, which describes how an update on $\chi_u$ propagates through the parameter space to affect the logits of $\chi_o$. It is sequence-aware due to token dependencies in the logits.

- $G_t = \nabla_z \mathcal{L}(\theta_t; \chi_u)$: The **Loss Residual**, which is the gradient of the loss in the logit space and dictates the direction and magnitude of the desired logit adjustment.

For a sufficiently small learning rate $\eta$, the $\mathcal{O}(\eta^2)$ term is negligible. This decomposition is consistent with the diagnosis of the squeezing effect (§2.1) presented in §2.2. $\qquad\square$

## I.2 DERIVATION OF CONSTRAINED SFT AND SMOOTHED LOSS (EQS. 3–4)

In Stage 1 (§3), we formulate the training objective as a constrained optimization problem:

$$\min_\theta \mathbb{E}_{(x,y^+)\sim\mathcal{D}}[-\log\pi_\theta(y^+|x)] \quad \text{s.t.} \quad \mathbb{E}_{(x,y^-)\sim\mathcal{D}}[-\log\pi_\theta(y^-|x)] \geq C.$$

**Derivation.** To solve this problem, we first form the Lagrangian, which incorporates the objective function and the inequality constraint:

$$\mathcal{L}(\theta,\lambda) = \mathbb{E}[-\log\pi_\theta(y^+|x)] + \lambda(C - \mathbb{E}[-\log\pi_\theta(y^-|x)]),$$

where $\lambda \geq 0$ is the Lagrange multiplier.

In practice, optimizing the exact Lagrangian over the full dataset expectation is intractable. We therefore adopt a more practical approach by creating a soft penalty formulation that can be optimized stochastically with mini-batches. We approximate the expectations with mini-batch averages and use the ReLU function to create a penalty that activates only when the constraint is violated (i.e., when $C - \mathbb{E}_{\text{batch}}[-\log\pi_\theta(y^-|x)] > 0$). This leads to the smoothed SFT loss:

$$\mathcal{L}_{\text{SFT-C}} = \mathbb{E}_{\text{batch}}[-\log\pi_\theta(y^+|x)] + \lambda \cdot \text{ReLU}\big(C - \mathbb{E}_{\text{batch}}[-\log\pi_\theta(y^-|x)]\big).$$

This expression matches Eq. 4. It ensures a bounded Negative Log-Likelihood (NLL) on negative samples, thereby smoothing the loss landscape and stabilizing the Belief Geometry term $A_t$ in preparation for Stage 2 (as specified in Algorithm 1, Step 5). □

## I.3 DERIVATION OF VANILLA DPO GRADIENT (EQ. 5)

The DPO loss function is given by:

$$\mathcal{L}_{\text{DPO}} = -\log\sigma\big(\beta(\Delta_w - \Delta_l)\big),$$

with $\Delta_{w/l} = \log\pi_\theta(y_{w/l}|x) - \log\pi_{\text{ref}}(y_{w/l}|x)$.

**Derivation.** Let $m = \beta(\Delta_w - \Delta_l)$ be the margin and $a = \sigma(m)$. We first compute the derivative of the loss with respect to the margin $m$. Using the derivative of the logarithm, $\frac{d}{dx}(-\log x) = -\frac{1}{x}$, and the derivative of the sigmoid function, $\frac{d}{dx}\sigma(x) = \sigma(x)(1 - \sigma(x))$:

$$\frac{\partial\mathcal{L}_{\text{DPO}}}{\partial m} = -\frac{1}{\sigma(m)} \cdot \frac{\partial\sigma(m)}{\partial m} = -\frac{1}{\sigma(m)} \cdot \sigma(m)(1 - \sigma(m)) = -(1 - a).$$

Next, we apply the chain rule to find the gradient with respect to the logits $z$:

$$\nabla_z\mathcal{L}_{\text{DPO}} = \frac{\partial\mathcal{L}_{\text{DPO}}}{\partial m} \cdot \nabla_z m = -(1 - a) \cdot \beta(\nabla_z\Delta_w - \nabla_z\Delta_l).$$

Since $\pi_{\text{ref}}$ is a fixed reference model, its derivative with respect to the active model's parameters (and thus logits $z$) is zero. Therefore, $\nabla_z\Delta_{w/l} = \nabla_z\log\pi_\theta(y_{w/l}|x)$. We define this term as $g_{w/l} = \nabla_z\log\pi_\theta(y_{w/l}|x)$, which for an autoregressive model simplifies to $\pi_\theta(\cdot|x) - y_{w/l}$ (where $y_{w/l}$ is a one-hot representation).

Substituting this back gives the final expression for the loss residual:

$$\nabla_z\mathcal{L}_{\text{DPO}} = \beta(1 - a)(g_w - g_l),$$

which matches Eq. 5. This form clearly highlights the role of the loser gradient term $g_l$ in the squeezing effect, as discussed in §2.2. □

## I.4 DERIVATION OF CW-DPO LOSS AND ASYMMETRIC GRADIENT (EQS. 7–8)

The CW-DPO loss function is defined as:
$$\mathcal{L}_{\text{CW-DPO}} = -\log \sigma(\beta \left(\Delta_w - w_c \cdot \Delta_l\right)),$$

with the cooling weight $w_c = \sigma\left(\frac{\bar{\ell}_\theta(y_l|\chi) - \ell_{\text{floor}}}{\tau}\right)$.

**Derivation.** Let $m' = \beta(\Delta_w - w_c\Delta_l)$ and $a' = \sigma(m')$. Following the same initial step as in the vanilla DPO derivation, we have:
$$\frac{\partial \mathcal{L}_{\text{CW-DPO}}}{\partial m'} = -(1 - a').$$

We then apply the chain rule. A key simplification in our derivation is to treat the cooling weight $w_c$ as a locally constant value during the gradient computation. This means we ignore the gradient flow through $w_c$ via its dependence on $\bar{\ell}_\theta$, which avoids computing complex second-order derivatives while still allowing $w_c$ to function as an adaptive modulator of the gradient magnitude.
$$\nabla_z \mathcal{L}_{\text{CW-DPO}} = \frac{\partial \mathcal{L}_{\text{CW-DPO}}}{\partial m'} \cdot \nabla_z m' \approx -(1 - a') \cdot \beta(\nabla_z \Delta_w - w_c \nabla_z \Delta_l).$$

Substituting the definition $g_{w/l} = \nabla_z \Delta_{w/l}$:
$$\nabla_z \mathcal{L}_{\text{CW-DPO}} = \beta(1 - a')\left[(\pi_\theta(\cdot|x) - y_w) - w_c(\pi_\theta(\cdot|x) - y_l)\right],$$

which yields Eq. 8. This result shows the core mechanism of our method: the cooling weight $w_c$ asymmetrically modulates only the gradient contribution from the loser sample, $g_l$, providing precise control over destabilizing gradients, as implemented in Algorithm 1, Step 11. □

## I.5 DISCUSSION AND FORMAL ANALYSIS OF REMARK 1: INSUFFICIENCY OF DPO'S IMPLICIT REGULARIZATION

**Remark 1 (Restated).** DPO's implicit regularization via the $\beta(1 - a)$ term falters for moderately easy negatives, perpetuating the squeezing effect.

**Analysis.** The effectiveness of DPO's gradient regularization hinges entirely on the factor $(1 - a)$.

- **For extremely easy negatives:** When a loser sample $y_l$ is confidently rejected, its log-probability $\log \pi_\theta(y_l|x) \to -\infty$, causing $\Delta_l \to -\infty$. This drives the margin $(\Delta_w - \Delta_l) \to +\infty$, and consequently, the sigmoid activation $a = \sigma(\beta(\Delta_w - \Delta_l)) \to 1$. In this asymptotic limit, the regularization term $(1 - a) \to 0$, effectively nullifying the gradient.
- **For moderately easy negatives (the "vulnerable region"):** In practice, the training set contains numerous negatives that are easy but not infinitely so. For these samples, $\log \pi_\theta(y_l)$ is low (e.g., in the range $[-10, -5]$) but finite. With typical values of $\beta$ (e.g., $[0.1, 1.0]$), the activation $a$ will be very close to 1 but not exactly 1 (e.g., in $[0.8, 0.99]$). This results in a regularization factor $\beta(1 - a)$ that is small but non-zero (e.g., in $[0.01\beta, 0.2\beta]$).

The problem arises because the raw gradient of the loser term, $g_l$, can be large and noisy. Multiplying this large, noisy gradient by a small but non-zero scalar still results in a non-negligible, noisy update signal. Formally, the variance of this gradient component, $\text{Var}(G_t^l) \propto (\beta(1 - a))^2 \cdot \text{Var}(g_l)$, remains significant and continues to introduce instability into the optimization process.

In contrast, the CW-DPO mechanism is more robust. For any negative sample where the average log-probability falls below the threshold, $\bar{\ell}_\theta(y_l|\chi) < \ell_{\text{floor}}$, our cooling weight $w_c$ is driven towards 0. The resulting variance of the cooled gradient, $\text{Var}(G_t^{\text{CW}}) \propto w_c^2(\beta(1 - a'))^2 \cdot \text{Var}(g_l)$, is thus effectively suppressed to zero. This acts as a much stronger and more reliable gating mechanism than DPO's soft, asymptotic regularization, directly addressing the core instability identified in §2.2. □

## STATEMENT ON THE USE OF AI ASSISTANCE

This manuscript was written entirely by the authors. A Large Language Model (LLM) was used only as a language assistant to check for grammar errors and improve clarity and readability.

