# OpenReview forum: "Learning Dynamics of VLM Finetuning: Cooling-Weighted DPO with Mixed Negatives"
_ICLR.cc/2026/Conference — ICLR 2026 Conference Withdrawn Submission_

### Official Review · Reviewer_CJZP · 2025-10-17

**Soundness:** 2
**Presentation:** 3
**Contribution:** 2
**Rating:** 4
**Confidence:** 4

**Summary:**

To address the known “squeezing effect” in preference-based learning, where gradients from easy negatives dominate updates and destabilize optimization, this paper introduces a weight term that suppresses uninformative gradients from easy negatives in the prefrence optimization phase. The proposed method has been evaluated on 3 image captioning benchmarks (COCO, Flickr30k, NoCaps), and 2 multi-task evaluation benchmarks (MMMU, MMBench), showing improved stability and performance.

**Strengths:**

1. Proposes a method that addresses an important challenge in preference-based learning (the squeezing effect), which causes instability in VLM alignment.
2. Shows good empirical performance in CIDEr score and other metrics across multiple benchmarks.
3. Provides an ablation that validates the significance of each component of the proposed method

**Weaknesses:**

1. The need for “performing supervised finetuning with gentle negatives: low-weight smoothed supervision that regularizes the base policy and curbs overconfidence without explicit penalties.” is weakly motivated. The paper emphasizes that easy negatives introduce uninformative gradients that destabilize learning, yet Stage 1 deliberately incorporates them during supervised finetuning, while Stage 2 explicitly suppresses their gradients. Why is it beneficial to include easy negatives early, only to down-weight them later? This design choice appears inconsistent with standard practice in preference-based learning (e.g., DPO, RLHF), where SFT typically focuses solely on positive samples.
2. The paper proposes to use “a cooling weight computed from the model’s average token log-probability on each negative” in the second stage to suppress easy negatives while preserving hard negatives. However, the method also mixes on-policy and dataset negatives. Typically, on-policy negatives are harder, while dataset negatives are easier because they have been frequently seen during training. I understand that dataset negatives are needed to prevent the model from being overfitted to on-policy negatives, but why do we need to suppress the easy negatives (or dataset negatives) then? The easy negatives that fall below the hyperparameter “l_floor” still might contribute in stabilizing training, so should might not be removed during DPO update.
3. The choice of the hyperparameter l_floor appears to play a critical role, as it defines the boundary between easy and hard negatives and thus controls the strength of gradient suppression. How sensitive is CW-DPO to this parameter? In particular, how robust is the method to variations of l_floor beyond using the mean log-probability value reported in the paper?
4. The paper states that “to ensure robustness, all reported results are averaged over five independent runs.” While I appreciate the effort to provide statistically reliable results, it would strengthen the paper to also report the standard error across these trials.
5. What are the possible limitations of this work beyond sensitivity to threshold values (l_floor, C)?
6. it would be better to leave some space between Figure4 and Line 340.

**Questions:**

Please address questions in the weaknesses section.

---

### Official Review · Reviewer_cpJ8 · 2025-10-27

**Soundness:** 2
**Presentation:** 3
**Contribution:** 2
**Rating:** 4
**Confidence:** 4

**Summary:**

This paper proposes an improved DPO-based method, Cooling-Weighted DPO (CW-DPO), to address instability issues in preference-based fine-tuning of vision-language models (VLMs). The authors first provide a theoretical analysis of the instability sources during VLM fine-tuning by decomposing the DPO gradient. Based on this analysis, they introduce a two-stage optimization framework consisting of Smooth SFT and CW-DPO. Extensive experiments on three image captioning benchmarks and two multimodal LLM evaluation benchmarks demonstrate the effectiveness of the proposed approach.

**Strengths:**

1. The paper is clearly presented and easy to follow. The theoretical analysis and experimental sections are well-explained, and the diagram in Figure 1 effectively illustrates the key concepts.

2. The proposed method achieves state-of-the-art performance across multiple vision–language benchmarks, outperforming other DPO-based approaches on COCO Test, Flickr30k Test, NoCaps Val, MMMU, and MMBench.

**Weaknesses:**

1. **Lack of empirical validation of theoretical claims**. While the paper provides a plausible dynamics-inspired motivation for the proposed two-stage training scheme, the experiments remain largely phenomenological. There is no direct evidence that Stage-1 indeed smooths the belief geometry (e.g., curvature or gradient correlation) or that Stage-2 explicitly regulates the loss residual as suggested by the theoretical analysis. For example, Figure 1 only compares training loss and average entropy, which does not directly validate the stabilization of belief geometry in Eq. 2. Similarly, the results mainly show distribution shift distance and generation quality, without demonstrating how the proposed stages contribute to belief regularization. It is recommended to include visualization or quantitative analyses that explicitly support these theoretical claims.

2. **Weak empirical comparison and ablation**. The empirical comparison lacks sufficient depth. While the dynamics-inspired two-stage motivation is interesting, the paper should better isolate the contributions of each stage in practice. The baselines in Table 1 do not adopt a two-stage optimization procedure, so it remains unclear how much of the performance gain arises from the Stage-1 (Smooth-SFT) design or from Stage-2 (CW-DPO). It would strengthen the paper to apply a similar two-stage optimization to other baselines for fair comparison and to demonstrate the general effectiveness of the proposed training dynamics.

3. **Limited extensibility and excessive hyperparameters**. The proposed method introduces several additional hyperparameters, $\lambda$, $C$, $ℓ_{floor}$, and $\tau$, which may limit its practical applicability and generalization to other tasks. The dependence on fine-tuning these parameters could hinder the method’s scalability and extendibility to broader preference-learning settings.

4. **Absence of human evaluation**. As this is a preference-based learning paper, the absence of any human evaluation is a significant omission. Including human preference assessments or human-aligned metrics would provide a more comprehensive validation of the proposed approach and its real-world relevance.

**Questions:**

1. In Figure 2, are the LoRA parameters fine-tuned in Stage-1 kept fixed during Stage-2? If so, it would be helpful to explain the rationale behind this design choice.

2. Several minor typos need correction, such as “our CW-DPO” in line 208 and the caption of Figure 2.

---

### Official Review · Reviewer_6LME · 2025-10-30

**Soundness:** 3
**Presentation:** 3
**Contribution:** 4
**Rating:** 6
**Confidence:** 3

**Summary:**

The paper introduces Cooling-Weighted DPO (CW-DPO), a two-stage method designed to address instability in VLM fine-tuning, specifically the "squeezing effect" that leads to uninformative gradients and unstable optimization. CW-DPO utilizes constrained SFT for loss landscape smoothing and competence-aware cooling weights to asymmetrically and adaptively suppress easy negatives.

**Strengths:**

1.The theoretical analysis is comprehensive, and the theoretical foundation is solid.
2.Some curves plot the mean and standard deviation.
3.The performance improvement of CW-DPO over Qwen2.5-VL is significantly better than other DPO variants.

**Weaknesses:**

1. Lack of implementation details in the main text, which makes it difficult to follow, such as the learning rate settings and device configuration.
2. No details on LoRA parameters (such as rank) or the layers being set (e.g., W_q, W_k, W_v in the attention mechanism).

**Questions:**

1. Is there a connection between this work and [1]? I would like to understand the differences in learning dynamics between LLMs and VLMs. I hope to compare the methods from [1] with this work either theoretically or experimentally.
2. Please supplement the experimental details mentioned in the weaknesses above.
3. The LoRA plot in Figure 2 feels a bit strange. Normally, LoRA should be embedded within Qwen2.5-VL. Could you explain the reasoning behind this particular plotting method?
[1] Learning Dynamics of LLM Finetuning

---

### Official Review · Reviewer_HBW6 · 2025-10-31

**Soundness:** 1
**Presentation:** 1
**Contribution:** 2
**Rating:** 2
**Confidence:** 5

**Summary:**

This paper introduces a two-stage VLM fine-tuning.  In the first stage, the model is trained with both chosen and rejected responses using SFT. The rejected responses are optimized with strong Anchored Preference Optimization [1,2], which stabilizes training. The second stage aims to mitigate the squeezing effect, where training easy rejected responses (those the model already confidently recognizes as wrong) with DPO causes overfitting. The proposed method, Cooling Weight DPO, is designed to apply weaker learning signals to easy rejected responses and stronger signals to hard ones. The approach is mainly evaluated on single-sentence captioning tasks such as COCO, Flickr, and NoCap.

In the paper, the chosen and rejected responses are referred to as winner and loser samples, or as positive and negative samples.

[1] mDPO: Conditional preference optimization for multimodal large language
models EMNLP 2024

[2] OPA-DPO: Mitigating Hallucinations in Large Vision-Language Models via DPO:
On-Policy Data Hold the Key CVPR 2025

**Strengths:**

1. The paper clearly defines the squeezing effect and highlights the need to apply an idea similar to Label Smoothing [3] in VLM training.
2. Their methods are well explained with detailed mathematical formulations, and Figure 2 provides a clear illustration.
3. The idea of applying SFT to rejected responses is novel, and the motivation for using different learning strengths for hard and easy rejected responses in DPO is well presented.

[3] When Does Label Smoothing Help? NeurIPS 2019

**Weaknesses:**

1. There is significant room for improvement in their presentation. It would be helpful if the authors made their presentation more reader-friendly and approachable. The reviewer lists below the concerns for the Abstract and Introduction and believes that a similar level of revision is needed in the later sections as well. It is hoped that the authors will find the following points helpful in improving the paper.
   1. Line 12: The transition in the sentence flow is abrupt. `We recast` → `To address this problem, we recast ~`
   2. Line 14: The phrase `,a two-stage` gives the impression that `Learning-dynamics-aware optimization` and `Cooling Weighted DPO` correspond to Stage 1 and Stage 2 respectively, but this is not the case.
   3. Line 14, Line 50: The expressions `explicitly models and exploits the training trajectory` and `to explicitly models and harness how the model's beliefs evolve` sound unclear, making them difficult to understand.
   4. Line 15: The use of `:` seems unnecessary. The phrase `supervised finetuning with gentle negatives` should be followed by a clear explanation, but instead, another complex expression, `low-weight smoothed supervision`, appears.
   5. Line 19: Why is `In practice` included? The phrase `Through this process, we achieve` is recommended.
   6. Lines 19–22, 52, 74: The following phrases interrupt the smooth reading experience for readers. Please note that a good paper presents ideas accessibly while still offering useful insights. `on-policy negatives`, `dataset negatives`, `contrast freshness`, `first-class signals`, `intricate learning dynamics` (since the abstract introduced `learning-dynamics-aware optimization` in bold as the method name, it would be better to use the same phrasing in the introduction), `a competence-aware preference optimization`, and `binary judgments, and open-ended tasks`.
   7. Abstract: The excessive use of bold text makes it difficult to read.
   8. Line 25: The use of `isolate` feels awkward. How about using `show` or `demonstrate` instead?
   9. Line 28: `simple, general` → `simple and general` is grammatically more accurate.
   10. Line 28: In addition, the phrase `a simple, general principle` sounds unnatural in this context. Would `an effective recipe` be a better fit?
   11. Introduction: It consists of only two paragraphs. Could you consider splitting it further? For example, the first part could cover (1) SFT and DPO in VLM training, and the second part could discuss (2) the challenges of DPO training.
   12. Line 38: The phrase `necessitating ~` sets the reader up to expect a transition like `So, we present ~`. However, the sudden appearance of `Preference-based finetuning is ~` feels unnatural.
   13. Line 42: In `trivially incorrect or off-distribution`, it may not be clear. How about `trivially incorrect or slightly OOD samples`?
   14. Line 43: The patterned list `These gradients disrupt optimization, degrade calibration, and produce overconfident, peaky posteriors.` lacks depth in meaning. `These gradients lead to instability and overconfidence in VLMs.` would be better.
   15. Line 44: `produce overconfident, peaky posteriors` → `produce overconfident and peaky posteriors`. /
   16. Line 44: Some readers may find the appearance of `Off-policy` abrupt.
   17. Line 50: The explanation of `learning-dynamics–aware optimization` is insufficient, while the discussion of `Cooling-Weighted DPO` is overly detailed.
   18. Lines 56–69: The Introduction should convey high-level information, but it instead presents full mathematical formulas. Moreover, this part do not provide information about $\pi_{\theta}, L, \pi_{ref}, \hat{l}_{\theta}, \sigma$.
   19. Lines 64–65: In contrast, excessive information is provided. For example, the concepts of winner and loser were already explained in Line 57.
   20. Figure 2: The spacing between the main text and the caption is narrow.
2. It would be valuable for the authors to make additional efforts to enhance the reliability of their results.
   1. For example, the authors could evaluate their method on additional benchmarks, such as AMBER, Object Hal, and POPE in OPA-DPO [2] or GAQ, SQA, MathVista, CharQA, OCRBench, MMV, and RealWorldQA in Cambrian [5].
   2. It is notable that the benchmarks mentioned above do not require the OpenAI API and can be easily evaluated on Qwen-VL using LMMs-Eval [6].
      1. The reviewer believes that evaluating a strong model such as Qwen2.5 VL 72B on COCO, Flickr, and NoCap offers little meaningful insight. Specifically, these datasets contain short single sentence captions of about ten words and are not suitable for evaluating a large VLM like Qwen2.5 VL 72B. By analogy, it is like assessing college students by asking them to solve elementary school math problems, which is not an appropriate way to evaluate their abilities.
      2. Based on the reviewer’s experience, it is hard to believe that the model shows consistently strong results on the single sentence captioning tasks in Table 1, measured by BLEU, METEOR, CIDEr, and SPICE, while displaying balanced changes in the ablation results.
      3. This skepticism stems from the limitations of n-gram matching based evaluation metrics. The reviewer requests that the authors provide standard deviation results. It may be more valuable to evaluate the model on detailed captioning tasks, such as those in benchmarks [7,8]. However, given the scale and capability of Qwen2.5 VL 72B, even these benchmarks may offer limited insight.
      4. Datasets such as COCO, Flickr, and NoCaps are considered extremely small for training a model like Qwen2.5 VL 72B. Overfitting or underfitting may easily occur. How did the authors address these issues, or were they not observed?
   3. PPO and GRPO require completely different datasets from DPO. In particular, they need multiple alternative rejected responses and relative preferences among them. Were all such datasets constructed? Based on the reviewer’s experience, it is extremely difficult to apply PPO and GRPO directly and achieve better performance than the original Qwen2.5 VL 72B. Could the authors make additional efforts to enhance the reliability of their results.
   4. Could the authors provide the prompt used to generate the `minimally perturbed alternatives` with GPT 4o?
3. One main contributions is the cooling weight for rejected responses. However, despite the authors’ emphasis through several equations, its actual impact appears quite limited.
   1. In the table on page 16 of the Appendix, when $l_{floor} = -5$, most rejected responses should be considered hard, meaning that cooling should have little effect. If the ablation study results for $l_{floor}$ remain similar, does this not imply that the effect of cooling is minimal?
   2. In Table 2, the results for MMMU and MMBench also show little performance drop even when cooling is removed.



[4] Mitigating Hallucinations in Large Vision-Language Models via DPO: On-Policy Data Hold the Key CVPR 2025

[5] Cambrian-1: A Fully Open, Vision-Centric Exploration of Multimodal LLMs NeruIPS 2024

[6] LMMs-Eval: Accelerating the Development of Large Multimoal Models https://github.com/EvolvingLMMs-Lab/lmms-eval

[7] Benchmarking and Improving Detail Image Caption. ICLR 2025

[8] Mllm-as-a-judge: Assessing multimodal llmas-a-judge with vision-language benchmark.ICML 2024

**Questions:**

1. Please refer to the weaknesses mentioned above.
2. Could the authors clarify the meaning of the light blue and green rectangles in the lower subfigure of Figure 2, as well as the term *situation*? Beyond the assumed probabilities, it would be more helpful to indicate the corresponding cooling weights.
3. Could the authors confirm whether they followed the `Formatting Instructions for ICLR 2026` provided in the official LaTeX files? The guideline specifies a certain text box height, but the height in this submission appears to be approximately 7.8 inches.

```
## General formatting instructions
The text must be confined within a rectangle 5.5~inches (33~picas) wide and
9~inches (54~picas) long. The left margin is 1.5~inch (9~picas).

## Final instructions
Do not change any aspects of the formatting parameters in the style files.
In particular, do not modify the width or length of the rectangle the text
should fit into, and do not change font sizes (except perhaps in the
\textsc{References} section; see below).
```

---

### Official Review · Reviewer_AmZL · 2025-11-04

**Soundness:** 2
**Presentation:** 2
**Contribution:** 2
**Rating:** 2
**Confidence:** 2

**Summary:**

This paper proposes Cooling-Weighted DPO (CW-DPO) to reduce the influence of  wrong negatives in VLM training stage. It use gentle negatives with minor errors for supervised finetuning in the first stage. In the second stage, CW-DPO uses cooling weight to adjust the negative-sample gradient for more stable optimization.

**Strengths:**

-  CW-DPO outperforms all compared methods on image captioning benchmarks (COCO, Flickr30k, NoCaps) and multi-task evaluation benchmarks (MMMU, MMBench).

- CW-DPO uses cooling weight to adjust the negative-sample gradient for more stable optimization.

**Weaknesses:**

- The paper is overall hard to read. The figures lack clarity: in Figure 1, some text is too small to be legible, and certain symbols, such as y similar y^similar , are not clearly explained. Besides, the introduction contains large segments of mathematical formulas, and some symbols in these formulas are also not clearly explained in the introduction.

- This paper does not fully comply with the ICLR formatting requirements.

**Questions:**

- How to get negative data with minor errors?

---

### Note · Authors · 2025-11-12

I have read and agree with the venue's withdrawal policy on behalf of myself and my co-authors.